# Measurements of Nearshore Ocean-Surface Kinematics through Coherent Arrays of Free-Drifting Buoys

Edwin Rainville[1], Jim Thomson[1,*], Melissa Moulton[1,2,*], and Morteza Derakhti[1,*]

[1]University of Washington - Applied Physics Laboratory, Air-Sea Interaction, Seattle, WA 98105, USA
[2]National Center for Atmospheric Research, Boulder, CO 80305, USA
[*]these authors contributed equally to this work

**Correspondence:** Edwin Rainville (erainvil@uw.edu)

**Abstract.** Surface gravity wave breaking occurs along coastlines in complex spatial and temporal patterns that significantly impact erosion, scalar transport, and flooding. Numerical models are used to predict wave breaking and associated processes, but many models lack sufficient evaluation with observations. To fill the need for more nearshore wave measurements, we deployed coherent arrays of small-scale, free-drifting buoys named microSWIFTs. The microSWIFT is a small buoy equipped

with a GPS module to measure the buoy's position and horizontal velocities and Inertial Measurement Unit (IMU) to directly measure the buoy's rotation rates, accelerations, and heading. Measurements were collected over a 27-day field experiment in October 2021 at the US Army Corps of Engineers Field Research Facility in Duck, NC. The microSWIFTs were deployed as a series of coherent arrays, meaning they all sampled simultaneously with a common time reference, leading to a rich spatial and temporal dataset during each deployment. Measurements spanned offshore significant wave heights ranging from 0.5 to 3

meters and peak wave periods ranging from 5 to 15 seconds over the entire experiment.

    The completed dataset consists of 67 deployment files that contain 971 drift tracks that have all associated data. We use an Attitude and Heading Reference System (AHRS), 9-degrees-of-freedom Kalman filter to rotate the measured accelerations from the reference frame of the buoy to the Earth reference frame. We then use the corrected accelerations to compute the vertical velocity and sea surface elevation. We give example evaluations of wave spectral energy density estimates from individual

microSWIFTs compared with a nearby acoustic waves and currents (AWAC) sensor. A zero crossing algorithm is applied to each buoy time series of sea surface elevation to extract realizations of measured surface gravity waves, yielding 116,307 wave realizations throughout the experiment. We also compute significant wave height estimates from the aggregate wave realizations and compare these estimates with the nearby AWAC estimates. An example of spatial variability of cross-shore velocity and vertical acceleration is explored. Wave breaking events, detected by high-intensity vertical acceleration peaks, are explored, and

the cross-shore distribution of all breaking events detected in the experiment is shown. A total of 3,419 wave breaking events were detected across the entire experiment. These data are available at https://doi.org/10.5061/dryad.hx3ffbgk0 (Rainville et al., 2023) and will be used to investigate nearshore wave kinematics, transport of buoyant particles, and wave breaking processes.

# 1 Introduction

The ocean covers most of the surface of the Earth, and in 2002 about 41% of the people on Earth lived within 100 km of the coast; we expect this population has continued growing (Boehm et al., 2017; Martínez et al., 2007). We expect sea levels to rise and storm frequency and intensity to increase due to climate change, making coastlines more susceptible to flooding, infrastructure damage, and loss of life (Intergovernmental Panel on Climate Change, 2021). Under moderate greenhouse gas emission forcing scenarios, we predict approximately $990 billion in damages to US coastlines between now and the year 2100 due to storm surge and sea level rise (Neumann et al., 2015). Wave forcing is a significant component of the total storm surge that causes flooding in low-lying coastal areas (Bertin, 2016). As surface gravity waves propagate towards the shore, they also transport energy and momentum, which drives nearshore circulation and scalar transport (Svendsen, 2005). Understanding these processes is essential for proper coastal management. Coastal managers use operational-scale forecast models that predict nearshore wave transformation, circulation, and transport but do not resolve individual waves and instead use a spectral representation of the waves in models such as SWAN (Simulating Waves Nearshore, (Booij et al., 1996)) and WWIII (Wave Watch III, (Tolman, 1999)). Other wave models are wave-resolving, such as NHWAVE (Derakhti et al., 2016; Ma et al., 2012) and FUNWAVE (Kirby et al., 1998; Shi et al., 2012), but are computationally expensive and therefore are not used operationally. Since the operational models do not resolve individual waves, they do not resolve individual wave processes, such as wave breaking. Therefore, to represent these processes in the operational models, we must parameterize them. A dataset with both wave-resolved and wave-averaged measurements is useful to investigate the wave-averaged model parameterizations for individual wave processes, especially breaking.

Fixed sensors, such as acoustic waves and currents (AWAC) meters and bottom-mounted pressure gauges, are commonly used to measure ocean surface waves. The data products produced from these types of instruments include the energy density spectra from which bulk wave parameters such as significant wave height and mean wave period are computed at a specific location (Birch et al., 2004). These fixed sensors generally have robust statistics since they measure continuously in the same location. Remote sensing methods, such as stereo video techniques and thermal infrared imaging, among many others, are also used to measure the waves in the nearshore with reasonable accuracy (Benetazzo, 2006; Carini et al., 2015). While it can be challenging to deploy many fixed sensors and remote sensing systems, many field campaigns have been successful using these methods in the nearshore region (Guérin et al., 2018; Pezerat et al., 2022; Lavaud et al., 2022; Carini et al., 2015; Elgar et al., 2001). As a complement to the fixed sensors and remote sensing methods, wave buoys are another option for obtaining direct measurements of the surface kinematics in various sea states. Wave buoys can be either free-drifting or moored. Moored buoys are effectively Eulerian wave measurements, with some movement due to the scope of the mooring, while free-drifting wave buoys are closer to Lagrangian measurements but move as a result of the wind, currents, wave-induced drift (Stokes drift), and surfing on broken waves. Free-drifting buoys are essential for understanding how buoyant objects move in the nearshore (Spydell et al., 2007; Schmidt et al., 2003). Free-drifting buoys tend to move through the surf zone very quickly; prior studies have reported buoys reaching approximately 50 $cm\,s^{-1}$ as a mean drift velocity. The velocities can be even larger in the presence of breaking waves and bores (Schmidt et al., 2003; Deike et al., 2017).

Early wave buoys used measurements of heave, pitch, and roll to compute the scalar and directional energy spectra (Kuik et al., 1988). The next generation of wave buoys, including the SWIFT (Surface Wave Instrumentation Floats with Tracking) buoy, focused on using GPS velocity-based processing methods (Thomson, 2012). The GPS methods have facilitated smaller-scale and more cost-effective wave buoys with comparable accuracy to fixed platforms (Herbers et al., 2012). The GPS methods can be limited to measuring deep-water waves when using horizontal velocities and assuming circular wave orbits (Thomson et al., 2018). GPS-based drifters have also been used to investigate surf-zone dispersion and circulation patterns (Schmidt et al., 2003, 2005; Spydell et al., 2007).

There are now many small, GPS-based wave buoys in common usage. SWIFT buoys have been used to measure turbulence (Thomson, 2012; Thomson et al., 2016), wave-ice interactions (Voermans et al., 2019) and wave-current interactions (Zippel and Thomson, 2017). SWIFTs have also been used to quantify the breaking severity of individual waves (Brown et al., 2019). The company *SOFAR ocean* has since developed the Spotter buoy that uses a GPS-based wave measurement (Raghukumar et al., 2019). Many Spotter buoys are deployed worldwide to create a global network of wave measurements that can be assimilated into a global wave model and thus assist industries reliant on accurate forecasts of waves.

While buoys have inherent challenges in measuring nearshore waves, including distortion of surface elevation from accelerometer measurements (Magnusson et al., 1999) and inability to resolve second-order non-linearity (Forristall, 2000), they are one of the few tools that can be used to obtain direct measurements of the kinematics of the surface. Buoys also provide the most direct observation of buoyant objects in the nearshore region. In the following sections, we discuss the deployment of microSWIFTs as part of the During Nearshore Events Experiment (DUNEX) (Section 2.1), the development of the version 1 microSWIFT wave buoy (Section 2.2), the creation of a large dataset including raw and post-processed measurements (Section 3), and the utility of that dataset for studying nearshore wave processes (Section 4).

## 2 Data Collection - During Nearshore Event Experiment (DUNEX)

This project is part of a larger collaborative effort called DUNEX (During Nearshore Event Experiment) that is funded through the US Coastal Research Program (USCRP, https://uscoastalresearch.org). The overall goal of DUNEX is to use rapid-response or other event-focused measurements and models to improve understanding of coastal impacts during storm events. Historically, it has been difficult to make measurements during such events. As a part of DUNEX, a 27-day field experiment was held from October 3-30, 2021. During the field experiment, we measured the motion of small-scale, free-drifting microSWIFT buoys in the inner shelf and surf zone. The microSWIFTs move with the free surface, thus providing measurements of surface kinematics. The following subsections will describe the data collection from DUNEX, including the field experiment, microSWIFT development, and data processing.

### 2.1 Field Experiment Site and Conditions - US Army Corps of Engineers Field Research Facility (FRF)

The experiment was conducted at the US Army Corps of Engineers (US-ACE) Field Research Facility (FRF) in Duck, North Carolina, USA (Figure 1). This site has a long history of being the focus of coastal dynamics experiments and is a relatively

well-understood sandy beach environment (Elgar et al., 1997; Gallagher et al., 1998; Feddersen et al., 1998; Elgar et al., 2001; Raubenheimer et al., 1996). The FRF maintains long-term observations via fixed in situ instrumentation, regular bathymetric surveys, remote sensing cameras, and LiDAR measurements. The FRF has an established local Cartesian coordinate system with the mean shoreline position located at approximately $x = 100$ meters, increasing offshore, and the location of the pier in the middle of the study site at approximately $y = 500$ meters, increasing northward (Figure 1, panel (b)). The bathymetry typically includes a large shore-perpendicular channel at approximately 500 meters in the alongshore direction underneath the pier due to scour. During the October 2021 study time period, a long shore-parallel sandbar at approximately 200 meters in the cross-shore direction was present (Figure 1, panel (c)). Previous studies suggest the bathymetry at this field site changes on timescales of hours to days (Ruessink et al., 2001).

A cross-shore transect of instruments (near $y = 900$ meters) is deployed and maintained by the FRF and includes several sensors in and near the surf zone that are relevant to this study (locations shown in Figure 1, panel (b)). The FRF completes all data processing for these instruments, and the data products produced are publicly available (data available at https://frfdataportal.erdc.dren.mil/). This sensor array includes a Nortek Acoustic Waves and Currents (AWAC) sensor in 4.5 meters mean water depth, another AWAC in 6 meters mean depth, and an array of pressure gauges in 8 meters depth used to estimate wave-directional spectra (Figure 1, Panel (b)). The data from these sensors are processed by the FRF staff to produce estimates of the bulk parameters of significant wave height, mean wave period, and mean wave direction for the duration of the field experiment (Figure 2) along with many other wave and current data products. During the experiment, significant wave heights ranged from 0.5 to 3 meters, mean wave periods from 5 to 15 seconds, and mean wave directions from 20 to 120 degrees relative to true North (where the cross-shore normal direction is 71.8 degrees clockwise of true North).

## 2.2 microSWIFT Buoy Development and Deployments

The microSWIFT buoys are named after their predecessor, the SWIFT buoy (Thomson, 2012). The version described herein is version 1; the Thomson et al (CEJ, in revision) for a description of other versions. The electronics and sensors of the microSWIFT are housed inside a Nalgene-brand water bottle with a length of 21 cm and a diameter of 9 cm (Figure 3). The Nalgene water bottle has a standard lid, and an O-ring was added to the mouth of the bottle to reduce water intrusions. The bottle sits on its side in the water, giving a keel of 4.5 cm and a sail of 4.5 cm. The overall microSWIFT has a mass of 0.7 kg. It is powered by two rechargeable lithium-iron D-cell batteries and has an approximate lifespan of 48 hours under the current operating configuration. The instruments on board the microSWIFT are a GPS module and Inertial Measurement Unit (IMU). A Raspberry Pi Zero, a small microprocessor with a Raspian Linux operating system, controls the entire system. The microSWIFT also has an iridium modem (RockBLOCK 9603) onboard that sends processed data from the microSWIFT to a shore-side server. The Raspberry Pi Zero also has an SD card where all raw data is stored and downloaded from when the buoys are recovered. Each component of the microSWIFT is soldered directly onto a custom circuit board (Figure 3). This is version 1 of the microSWIFT.

All software for the microSWIFT is written in the Python computing language and is published on a public Github repository for open source access (https://github.com/SASlabgroup/microSWIFT). The flow of onboard software is shown in Figure 4.

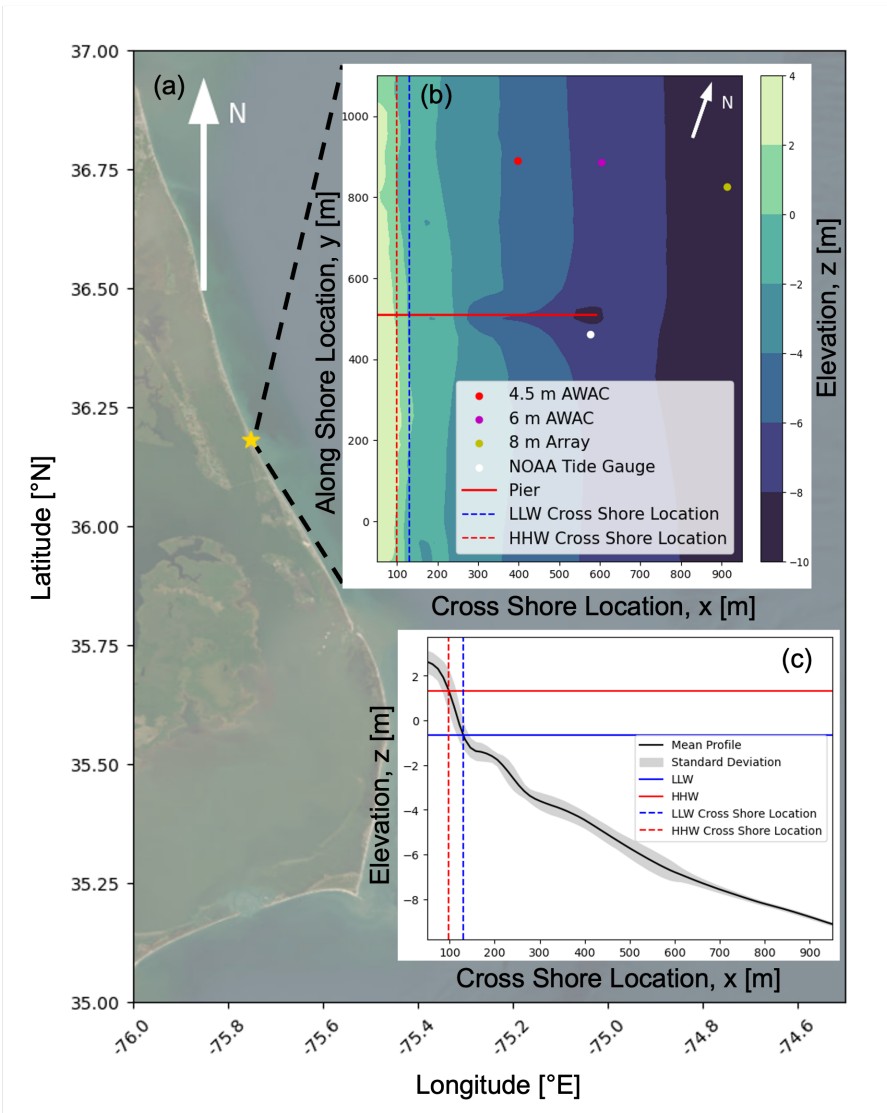

**Figure 1.** (a) Aerial imagery of the Outer Banks of North Carolina, US, where the gold star is the location of the US Army Corps of Engineers - Field Research Facility (FRF) (© Source: Esri, Maxar, Earthstar Geographics, and the GIS User Community). Panel (b) shows the bathymetry contours at the field site from October 21st, 2021, relative to the NAVD88 datum and locations of fixed instrumentation (Data provided by USACE, Field Research Facility, https://frfdataportal.erdc.dren.mil/). Panel (c) shows the average cross-shore profile of the bathymetry with one standard deviation above and below the average. The higher high water (HHW) and lower low water(LLW) levels measured during the experiment are also shown.

The microSWIFT is controlled by one primary function named *microSWIFT.py* that controls all other functions. When the microSWIFT boots up, a service script named *microSWIFT.service* runs and starts the main *microSWIFT.py* control function.

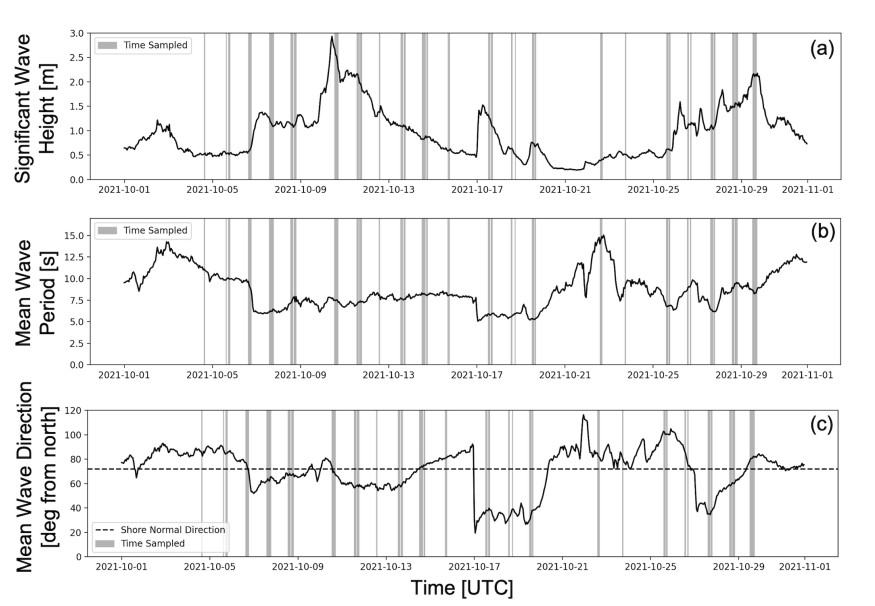

**Figure 2.** Conditions sampled during the main DUNEX experiment from the Field Research Facility 8 m array. Time series of the (a) significant wave height and (b) mean wave period, and (c) mean wave direction (Data provided by USACE, Field Research Facility, https: //frfdataportal.erdc.dren.mil/). The gray patches show the times that we deployed microSWIFT arrays.

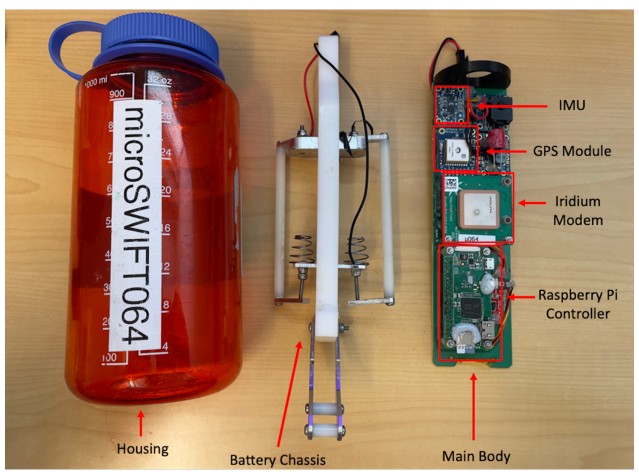

**Figure 3.** Layout of microSWIFT hardware components with the Nalgene water bottle housing on the far left, battery chassis in the middle, and electronics on the far right. The individual chipsets include a Raspberry Pi Zero as the main processor, a GPS module, an inertial measurement unit(IMU) that contains accelerometers, gyroscopes, and magnetometers, and an iridium modem.

As *microSWIFT.py* starts, it creates a log file where all functions onboard the microSWIFT are logged so the user can see if any errors are occurring or the instrument is working correctly. The microSWIFT central control is split into two windows, the record and process/send windows, with user-defined lengths based on universal coordinated time (UTC). Within the record window, the microSWIFT concurrently records GPS and IMU data and writes the data directly to a file. The microSWIFT enters the process/send window when the record window ends. Here, the microSWIFT reads in all of the recorded GPS velocities and

uses the algorithm *GPSwaves* described in Thomson (2012) and Thomson et al. (2018) to compute an estimate of the wave energy scalar spectrum, bulk parameters, last known location, and the average north-south and east-west velocities over the length of the last record window. These processed values are then packaged into a binary message sent through the iridium modem to a shore-side server where the data can be parsed and used. The *GPSwaves* algorithm is very effective for deep water waves; however, it uses an assumption of circular wave orbital velocities to estimate the scalar energy spectrum. The elliptical

orbits of shoaling waves in shallow water violate this assumption. The nonlinearity of shallow water waves and breaking waves further complicates the usage of GPS horizontal velocities to infer wave elevations. For nearshore applications, we developed a new processing method described in Section 3.

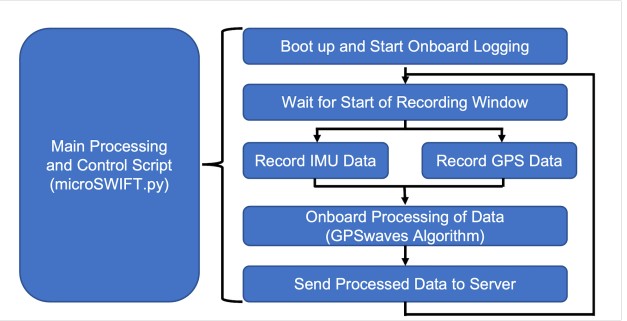

**Figure 4.** Flow of operations for software onboard the microSWIFT wave buoys.

As the microSWIFT drifts, the IMU and GPS sensors measure the motion of the bottle. The IMU measures accelerations, rotation rates, and magnetic heading in three orthogonal spatial dimensions at a rate of 12 Hz in the reference frame of the

140 buoy. The chipsets, sensitivities, and ranges of the accelerometer, gyroscope, and magnetometer are shown in Table 1. The GPS receiver is an MT3339 chipset that samples at a rate of 4 Hz and measures latitude, longitude, and horizontal velocities in the Earth reference frame. These measurements provide a comprehensive picture of how and where each microSWIFT moves in response to waves and surface currents.

Each microSWIFT provides detailed information about a single point in space and time. However, when deployed in large

numbers as coherent arrays, the microSWIFTs can be processed together to improve measurements in certain areas or explore spatial variability. The deployed coherent arrays ranged in size from two to fifty microSWIFTs. The microSWIFTs were deployed by throwing them from the pier, paddling them out on surfboards, dropping them from a helicopter, or providing them to local lifeguards who dropped them off a jetski. The microSWIFTs were retrieved when they eventually washed up on the beach or were chased down using surfboards, jet skis, and boats. To track the buoys in live time, we deployed 'shepherd'

| Sensor | Chipset | Sensitivity | Range | Average Noise Variance |
|---|---|---|---|---|
| 3-axis Linear Accelerometer | FXOS8700CQ | $0.244\ mg\ LSB^{-1}$ $0.488\ mg\ LSB^{-1}$ | $\pm 2g$ / $\pm 4g$ | $0.00004\ m\ s^{-2}$ |
| 3-axis Magnetometer | FXOS8700CQ | $0.1\ \mu T\ LSB^{-1}$ | $\pm\ 1200\ \mu T$ | $2\ \mu T$ |
| 3-axis Gyroscope | FXAS21002C | $15.625\ mdps\ LSB^{-1}$ | $\pm\ 500\ dps$ | $0.045\ dps$ |

**Table 1.** Inertial Measurement Unit sensor specifications for accelerometer, gyroscope, and magnetometer onboard each microSWIFT. Note that the dynamic range of the accelerometer was adjusted from 2g to 4g part-way through the field experiment on Mission 53 on October 23rd, 2021.

buoys which had the same hull and ballast as the microSWIFTs but had a live tracking GPS in them to track the current movement of the buoys as they drifted. Over the course of the experiment, 2,187 buoys were deployed, including the shepherd buoys, and only one was lost. We refer to each deployed array as a "mission." Drift tracks (location of each microSWIFT over the time of a mission) from the microSWIFTs on two example missions are shown in Figure 5. After data cleaning, the dataset consists of 67 missions spanning 27 days. The shoreline and surf zone edge estimates are added to panels (a) and (b) in Figure 5 to add context to the microSWIFT deployment. The shoreline and surf zone edge are both estimated using along shore averaged bathymetry (measured on October 21st, 2021 described in Figure 1) in combination with the mean water level from the NOAA tide gauge (location shown in Figure 1, panel (b)). For each mission, the mean water level during the deployment is added to the along shore bathymetry profile to give a cross shore depth profile during the mission. The shoreline is then estimated as the cross shore location where the depth during the mission equals zero on the along shore averaged profile. Waves are expected to begin breaking when the ratio of wave height $H_s$ to water depth $d$,

$$\gamma_s = \frac{H_s}{d}, \tag{1}$$

reaches a certain threshold. Using this definition of $\gamma_s$, the variable $H_s$ represents the offshore significant wave height, here measured from the 8 m-pressure gauge array (location in Figure 1, panel (c)), and the variable $d$ represents the water depth during the mission. Values of $\gamma_s$ from the Duck, NC field site have been observed to be between 0.4 and 0.8 (Sallenger Jr and Holman, 1985). Further studies have shown that within the inner surf zone at the Duck, NC field site $\gamma_s$ can reach as low as approximately 0.275 and as high as 0.375 at depths greater than 0.8 meters (Raubenheimer et al., 1996). Smaller values of $\gamma_s$ drive the breaking depth to deeper water, and larger values drive the breaking depth to shallower waters. From these observed values, we chose $\gamma_s = 0.35$ to provide a representative estimate of the surf zone edge location. These estimates are shown in both panels of Figure 5, and the same estimation method is used to add context to the analysis later on. The choice of $\gamma_s = 0.35$

is a traditionally low value but is used as a conservative estimate to include the outer surf zone where intermittent breaking is prevalent.

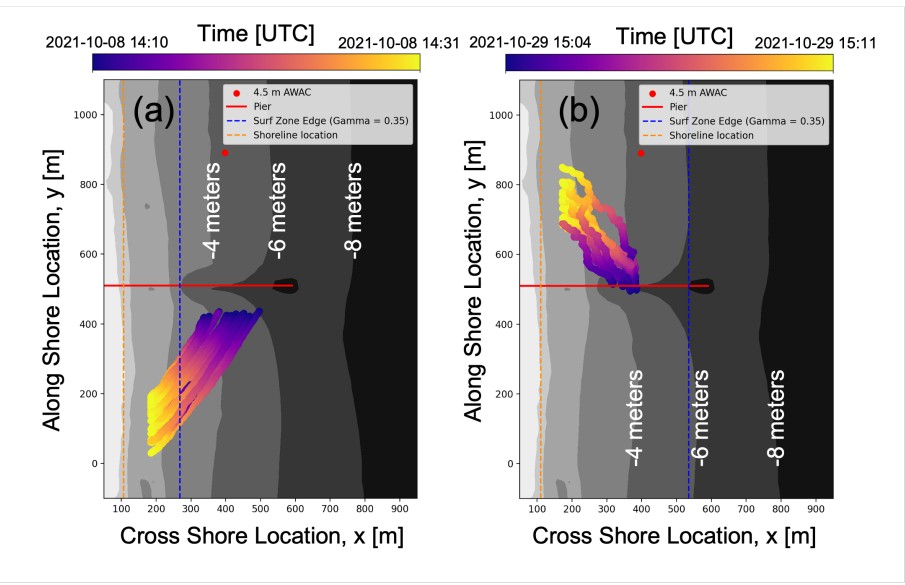

**Figure 5.** Example drift tracks(location of microSWIFTs over time) of microSWIFT arrays during a mission plotted over of the bathymetry digital elevation model shown in Figure 1 Panel (b). Panel (a) shows the drift tracks from mission 16, which has 19 microSWIFTs deployed, and Panel (b) shows the drift tracks from mission 79, which has 13 microSWIFTs deployed. Approximate shoreline and surf zone edges are shown for each mission.

## 3    Data Levels and Data Processing

We separate three levels of data as follows:

- Level 0: Text files of raw data from the GPS and IMU from each microSWIFT, organized by the mission number

- Level 1: GPS and IMU measurements stored in a netCDF file, data have been cleaned and interpolated onto the same time datum for each mission

- Level 2: IMU accelerations that have been corrected to the Earth frame of reference, with velocities and positions computed from these corrected accelerations

First, we download the Level 0 raw measurements from the microSWIFTs, organize them into folders for each mission, then
read the data from the text files. We then create a single time array with the mission's manually recorded start and end times. The start time is when all the microSWIFTs entered the water, and the end time is when all the microSWIFTs are out of the water. The time step in this time array matches the IMU sample rate of 12 Hz. We then match the IMU measurements to this

time array and linearly interpolate the GPS data to match the time array. Any gaps in the IMU measurements are filled with linear interpolation. Gaps in GPS measurements occur due to the buoys being overtopped or plunged underwater. These gaps are filled using linear interpolation but are generally minor. Schmidt et al. (2003) used similar GPS-based drifters and found 95% data return rates seaward of the surf zone and 75% data return rates within the surf zone but found linear interpolation was an appropriate method to fill the data gaps, so this methodology is followed.

## 3.1 Data Cleaning and Level 1 Data

We then clean these data using a combination of automated and manual methods. First, we create a spatial threshold to remove data points while a microSWIFT is on the beach. We create this threshold using a digital elevation model of the bathymetry at the FRF from October 21st, 2021 (elevation is relative to NAVD88) (Figure 1, Panel (b)) and the mean water level during each mission measured by a NOAA tide gauge (Location in Figure 1, Panel (b)). The mean water level is added to the bathymetry measurements to find the depth at each surveyed point during each mission. We find the furthest offshore dry point as the furthest offshore positive value. We then add a buffer of two additional meters to the furthest offshore dry point and set this location as the spatial threshold for that mission. We replaced any points that cross this beach threshold on the beach side with "NaN" values in the dataset. While there were changes in the bathymetry during the experiment, we are only using this survey to define an approximate location of the beach extent to do a bulk data cleaning and add some approximate context for these data. Further detailed analyses of other bathymetry surveys will be completed in future studies using this dataset. After this automated cleaning method, we manually inspected each data channel for any spurious points that were replaced with "NaN" values.

The recorded start and end times of the mission were also manually adjusted to reflect the times the microSWIFTs were actually in the water. Large spikes in acceleration at the beginning of a deployment tend to represent times when the start time was recorded too early and was adjusted to remove these spikes. Similarly, the microSWIFTs were sometimes picked up in the middle of the mission, e.g., during jetski-based deployments, and those times were manually removed as well. All data cleaning, including start and end time adjustment and individual point cleaning, is noted in Appendix B in Rainville (2022), and the cleaning notes are stored in the metadata of each netCDF and in the GitHub repository that contains all processing code (https://github.com/SASlabgroup/DUNEXMainExp). The IMU data is then despiked using a piecewise cubic Hermite interpolating polynomial (PCHIP) function, a shape-preserving interpolation scheme used to reduce overshoot oscillations and maintain continuity (Karim et al., 2014). Points with a value greater than four scaled median absolute deviations from the median are considered outliers and replaced using the PCHIP method. The cleaned and despiked dataset is considered the Level 1 data.

## 3.2 Level 2 Data

We use the gyroscope and magnetometer measurements to correct the accelerations from the body reference frame to the Earth reference frame using a 9 degrees-of-freedom indirect Kalman filter for IMU sensor fusion that is prepackaged in the MATLAB navigation toolbox (MATLAB Navigation Toolbox 2022b, https://www.mathworks.com/help/nav/multisensor-

positioning.html). A schematic representation and an example corrected signal are shown, respectively, in Panels (a) and (b) of Figure 6. The corrections to the acceleration are generally minor (see changes between uncorrected and corrected vertical acceleration in Figure 6, Panel (b)) but have a significant impact on the integrated signals and eventually computed energy spectra and bulk wave parameters. We then use a first-order Butterworth band-pass filter to remove low ($f < 0.05$ Hz) and high

($f > 0.5$ Hz) frequency noise outside of the gravity wave band from the signals. We then integrate the filtered acceleration signals via a time domain cumulative trapezoid method to velocities. The velocities are filtered again with the same filter to eliminate any spurious integration errors, then integrated to estimate positions, and finally filtered one last time to eliminate integration errors. The corrected and filtered accelerations, velocities, and positions are the Level 2 data.

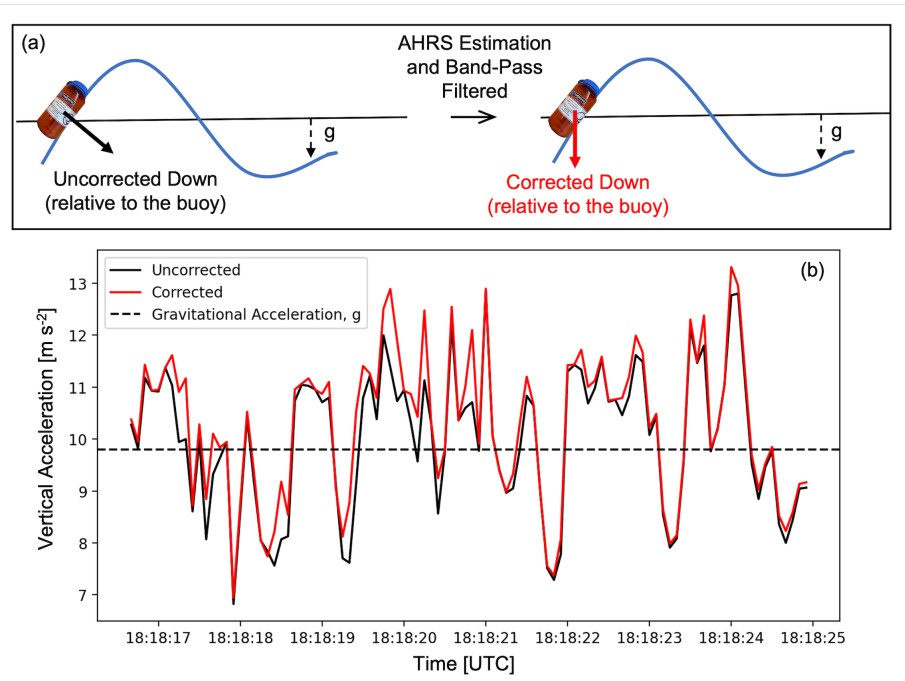

**Figure 6.** Panel (a) shows a schematic representation of the acceleration corrections from the body reference frame of the microSWIFT to the Earth reference frame through the use of the attitude and heading reference system (AHRS) estimation and band-pass filtering. Panel (b) shows an example portion of a signal to see how the vertical acceleration is corrected.

### 3.3 Spectral Exploration of microSWIFT Data

We assume that the main control of nearshore wave evolution is the local bathymetry, and therefore measurements in similar depths at the same time should be comparable in a spectral and statistical framework (Gomes et al., 2016). Spatial variability in the bathymetry can also lead to areas of convergence and divergence of wave energy through refraction, causing spatial variability of wave properties. These spatial differences can also be exaggerated through the blocking of obliquely incident waves by the pier leading to reduced wave energy in the 'shadow' as much as 50% reduced at 200 meters down-wave of the

pier (Elgar et al., 2001). We also expect measurements outside the surf zone to be more reliable for estimating wave properties since they are exposed to fewer breaking waves. Breaking waves tend to manifest as short bursts of high frequency and amplitude accelerations (Sinclair, 2014; Brown et al., 2019; Feddersen et al., 2023). Integrating these acceleration bursts can lead to spuriously large or nonphysical sea surface elevations; therefore, we expect the best agreement of wave measurements when the buoys are outside the surf zone or in the outer surf zone where breaking is more intermittent.

The 4.5-meter AWAC is currently at a bottom elevation of -4.8 meters relative to NAVD88 due to changes in the bathymetry since the instrument was initially deployed and named. In prior studies, AWAC sensors have been validated for use in the nearshore environment (Pedersen et al., 2007). Known limitations of the AWAC include excess noise at high frequencies and reduced response at low frequencies. To compute an energy spectral density estimate from an individual microSWIFT that we can compare to the 4.5-meter AWAC, we use data when an individual microSWIFT was at a location that corresponded to a bottom elevation between -4.3 and -5.3 ($\pm$ 0.5 meters around the current elevation of the AWAC) meters based on the local bathymetry measurements in Figure 1, Panel (b). As an example, mission 18 has four microSWIFTs that were between these depths for almost ten consecutive minutes as they drifted through the surf zone (Figure 7, Panels (a) and (b)).

The microSWIFT spectra are computed using Welch's method, with five-minute (3600 sample) Hanning windows and 50% overlap between adjacent windows. The energy in each of the five adjacent frequencies is band-averaged to improve the statistical robustness of each estimate. The equivalent degrees of freedom for each microSWIFT spectrum is 28. This is based on the ten-minute time series (7200 samples at a 12 Hz sampling rate) used for each spectral estimate with three five-minute windows (50% overlap). Each window contributes 2 degrees-of-freedom and band-averaging the five adjacent frequencies increases the effective degrees of freedom by a factor of five. Due to the 50% overlap of the Hanning windows, the equivalent degrees of freedom are reduced to 92% of the maximum degrees of freedom (Nuttall and Carter, 1980). Therefore, the equivalent degrees-of-freedom for the microSWIFT spectra is 28 (3 windows * 2 degrees-of-freedom * 5 frequency bands * 0.92 = 28). The AWAC measurements consist of a 34-minute record with a sample rate of 2 Hz, and spectra are computed with 13 50%-overlapping windows (512 points per window) and no band-averaging, leading to approximately 25 degrees-of-freedom, comparable to that of the microSWIFTs (Christou et al., 2011). The staff of the Field Research Facility process the AWAC data following the methods described by Earle et al. (1999). These spectral characteristics result in a frequency resolution of 0.016 Hz. Note, however, only the data products processed by the FRF are used in this study, and no processing of the raw AWAC data is done in this study. Each of the spectra computed from the microSWIFTs compares well with the spectra, and the averaged spectra also compare well with that reported from the 4.5 m AWAC (Figure 7, Panel (c)). The qualitative agreement of each microSWIFT spectra and the AWAC suggests that the measurements are useful for further investigating wave properties with the buoys. Future use of these data may investigate the estimation of directional spectra and directional moments, but they are not investigated in this study.

### 3.4 Zero-Crossing Exploration of microSWIFT Data

Since the buoys drift quickly through the surf zone, it is not always reasonable to compute spectra from each buoy since the buoys enter areas with active breaking, and as they move through different depths, the signal is not necessarily stationary. In

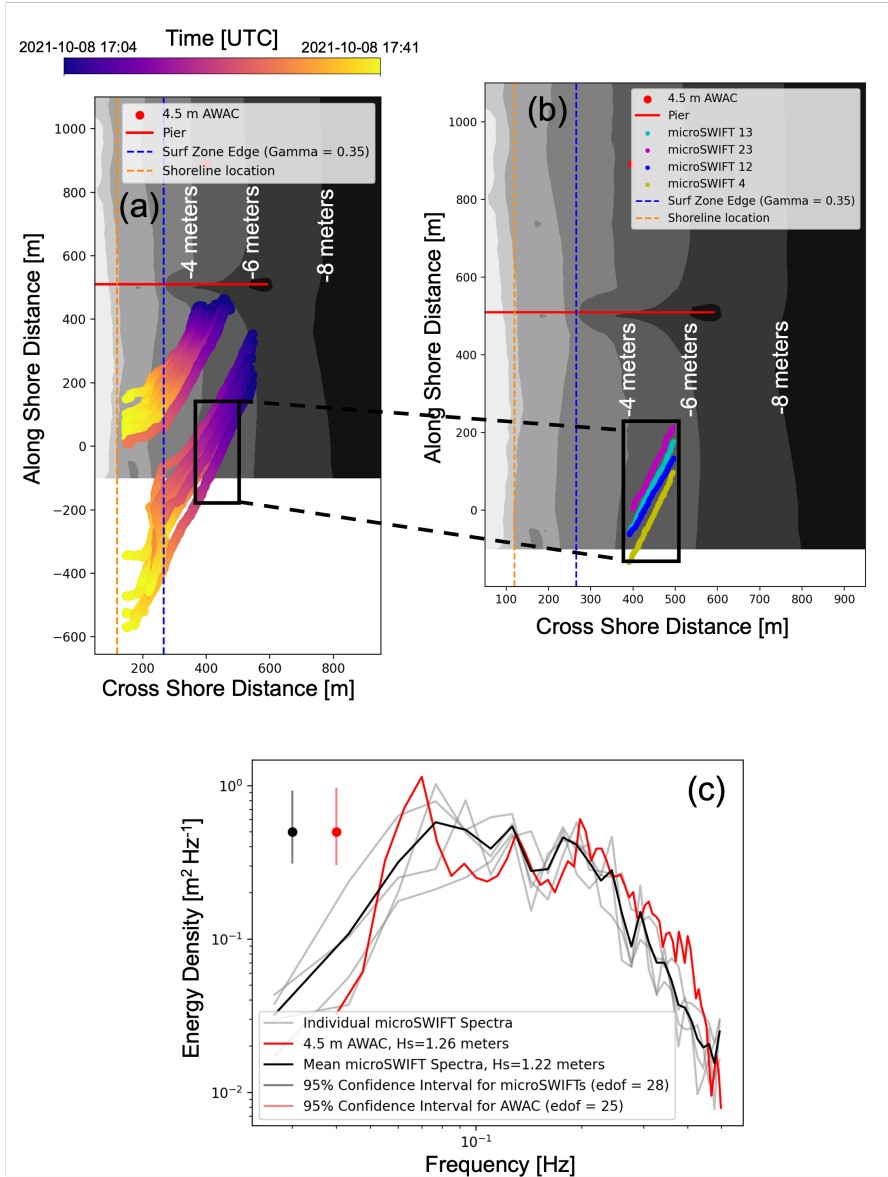

**Figure 7.** Comparisons of Panel (a) shows the drift tracks of the microSWIFTs from mission 18 plotted over the surveyed bathymetry DEM. Panel (b) shows a subset of the drift tracks where the bathymetry along each track is between -4.3 and -5.3 meters, and each microSWIFT is a different color. Panel (c) shows the spectra computed from a subset of the sea surface elevation time series for each microSWIFT. Significant wave heights are computed by numerically integrating the AWAC and averaged microSWIFT spectra.

this case, we use the arrays of microSWIFTs coherently to estimate the significant wave height. The measurements from each

buoy are combined using a zero-crossing algorithm, which identifies individual wave realizations along the drift track of each microSWIFT (Example processing of data using the zero-crossing approach is shown in Figure 8). Panel (a) shows the drift

tracks of each microSWIFT over time, while Panel (b) shows that each track is a different microSWIFT; each track is now a different color. Using the zero-crossing algorithm on each sea surface elevation time series, we define an individual wave realization as the data between two consecutive upward zero-crossings in elevation (Figure 8, panel (d)). The height of each
wave realization, from crest to trough, and the average location of the wave realization are also defined by this method. Since the microSWIFTs are spatially distributed in the nearshore and sampling simultaneously, some microSWIFTs will measure the same wave as it propagates past multiple buoys. We treat this like a physical 'sampling with replacement' method similar to Monte Carlo or Bootstrap simulation methods known as re-sampling techniques. These types of re-sampling techniques are used to improve the confidence in a statistical estimate from a finite amount of data (Thomson and Emery, 2014). In this case,
the finite data is the short period that the buoys sample an area, but multiple datasets from different microSWIFTs, occasionally containing measurements of the same wave, can help improve confidence in the statistics. The height of each wave realization is then aggregated, and the distribution of wave heights sampled during this mission is shown in Figure 8, Panel (e).

The distribution of wave heights follows a Rayleigh distribution as is expected for nearshore surface gravity waves (Thornton and Guza, 1983). The significant wave height is computed from aggregated wave height measurements outside the approximate
surf zone. The significant wave height is computed by first computing the root-mean-square of the wave heights and then multiplying by a factor of 1.414 to convert to significant wave height for a Rayleigh distribution (Dean and Dalrymple, 1991). An example of where this measure falls on the Rayleigh distribution is shown as a vertical line on the wave height distribution in Figure 8, Panel (e). By applying this processing to each mission in the experiment, we get a total of 116,307 wave realizations across the experiment. Using the locations of the wave realizations, the measured bathymetry, and the water level gauge, we
can approximate the depth of each wave realization across the experiment. Most wave realizations were on the south side of the pier between -2 and -6 meters in bottom elevation.

We evaluate this zero-crossing processing method by comparing significant wave height estimates from the microSWIFTs to the 4.5-meter AWAC, 6-meter AWAC, and 8-meter pressure sensor array (6-meter AWAC and 8-meter array estimates have been corrected for expected shoaling using linear wave theory). Like the 4.5-meter AWAC, the 6-meter AWAC has shifted
over time and is now at a bottom elevation of -6.83 meters. We first find, for each mission, all wave realizations that are located outside of the approximate surf zone. With this subset of waves, we compute the significant wave height as described previously. To calculate a significant wave height from one of these subsets of data, we require at least 30 wave realizations in the distribution. Thus, we do not compute a significant wave height for every mission. We compare the computed significant wave heights to those from the 4.5-meter AWAC, 6-meter AWAC, and 8-meter array (Figure 9).
The linear regression between the 4.5 m AWAC and microSWIFT array significant wave heights has a slope of 0.61 and an $R^2$ value of 0.74, showing a positive correlation between the two significant wave height estimates. This agreement is reasonable given that the microSWIFTs are measuring at a different alongshore location than the AWAC, although in similar water depths. We also expect that the microSWIFT arrays have more variability in their significant wave height estimates since the sampling windows are shorter than the AWAC, potentially over-representing or under-representing the largest and least
likely waves in the distribution. Further underestimation could be due to the microSWIFTs being within the 'shadow' of the pier. Being in the pier 'shadow' is defined here as missions when the average location of the microSWIFTs during a mission

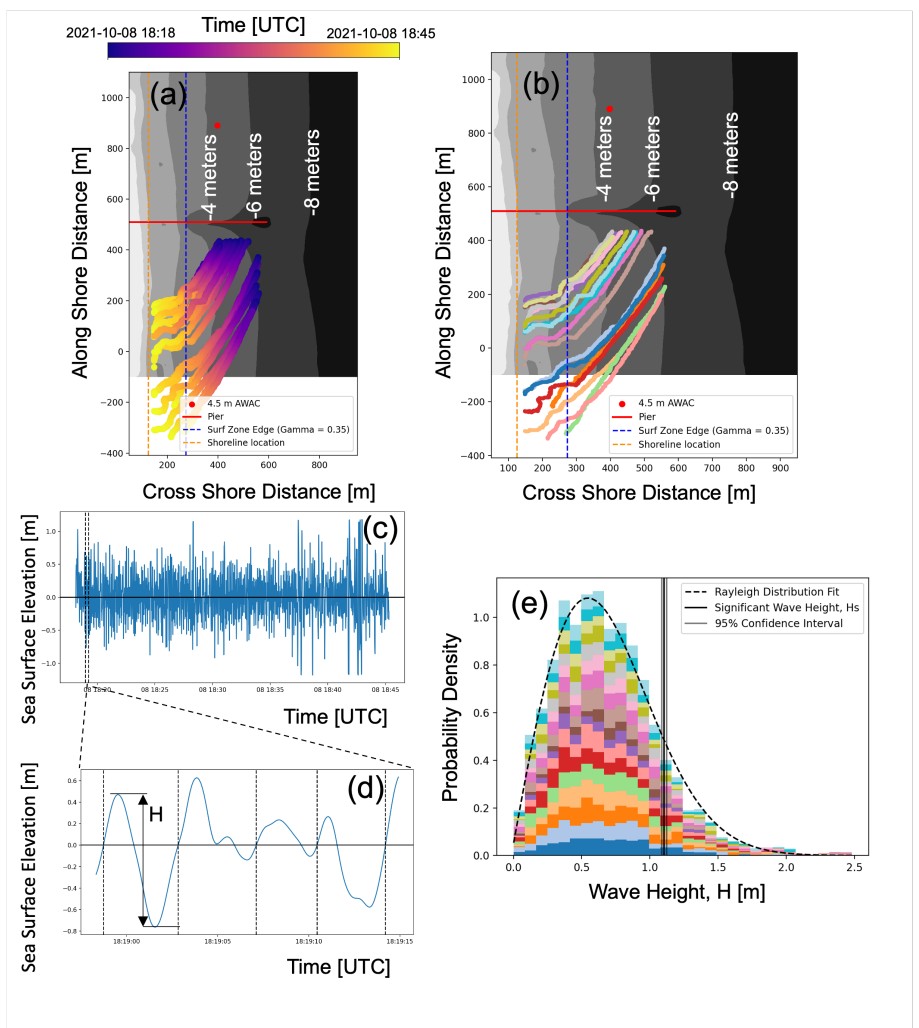

**Figure 8.** Example of steps in processing each mission. Panel (a) shows the drift tracks of the microSWIFTs from mission 19 plotted over the surveyed bathymetry DEM. Panel (b) shows the same drift tracks as Panel (a) but shows each microSWIFT as a different color. Panel (c) shows the time series of computed sea surface elevation, with one time series being highlighted as an example. Panel (d) is a zoomed-in portion of the overall time series showing the locations of zero crossings and how we define the height of an individual wave in a time series. Panel (e) is the probability density of all wave heights, seaward of the approximate surf zone edge, where the colors show the contribution from each microSWIFT with the corresponding color. The probability density distribution fits a Rayleigh distribution. The vertical line shows the computed significant wave height for this distribution and the 95% confidence interval of the estimate.

is within 200 meters of the pier, and waves are coming from the other side of the pier based on the mean wave direction from the 8-meter array (furthest offshore sensor). The significant wave height measurements from the 6-meter AWAC and 8-meter array are also adjusted to be in the same depths using linear wave theory and compared to show agreement between multiple

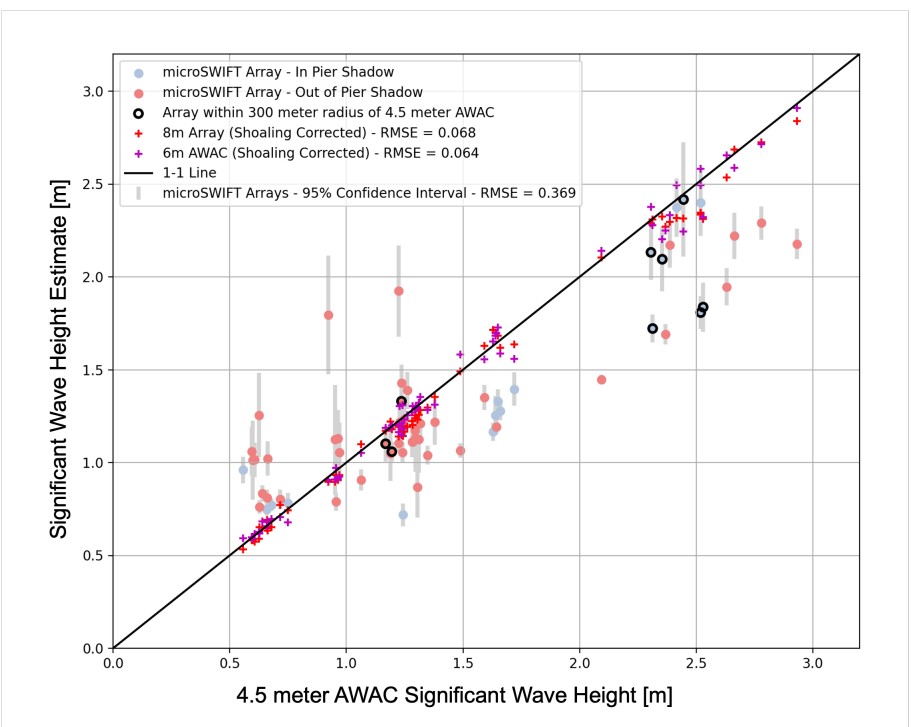

**Figure 9.** Comparison of the estimated significant wave heights from the microSWIFT arrays, 6-meter AWAC, and 8-meter pressure sensor array (6-meter AWAC and 8-meter array have been corrected for shoaling) to the estimates from the 4.5 m AWAC. While the microSWIFT arrays are not in the same water depth as the 4.5 m AWAC, we see that the microSWIFT arrays characterize the size of the waves with good comparison to the 4.5 m AWAC. The gray bars indicate 95% confidence intervals around each of the significant wave height estimates, computed using a bootstrap method from the distributions of wave heights. The colors of the estimates depict if the microSWIFT array is in the 'shadow' of the pier where we expect a reduction in wave energy. For significant wave heights greater than 2 meters, intermittent breaking may be occurring at the 4.5 meter isobath leading to worse agreement between the AWAC and microSWIFT measurements.

sensors for a more robust comparison. The agreement in significant wave height and scalar energy density spectra supports that the Level 2 data are useful for investigating wave spectra and statistics.

## 4    Data Use

The following section will describe some potential uses for this dataset. The level 1 data in this dataset consists of the cleaned and despiked GPS positions, GPS horizontal velocities, accelerations in the body frame of reference, rotation rates, and the magnetic heading of the buoys. These data channels contain information that can be used to investigate wave breaking motions and the transport of buoyant objects in the nearshore region under various conditions. The location of measurements made by the microSWIFTs during this experiment is shown in Figure 10. The orange dashed line shows the average location of the shoreline during the experiment. The closest-to-shore and furthest-off-shore surf zone edge estimates during the experiment

are shown and are based on the smallest and largest offshore significant wave heights during the experiment, respectively. This shows many measurements both outside and inside the surf zone. These measurements can help investigate buoyant particles' cross and along-shore transport under various forcing conditions. The microSWIFTs move with both the waves and the currents. They are buoyant, and thus they also tend to 'surf' on the broken waves. The buoys surfing can enhance the transport of these objects (Pizzo, 2017). This type of motion affects similar objects transported in the surf zone, such as large algae, e.g., Sargassum, a buoyant seaweed affecting coastlines in the south-eastern US (Webster and Linton, 2013). The mean surface currents, Stokes drift, and rip currents are resolved within transport models for surface-constrained particle motion (Moulton et al., 2023).

The process of 'surfing' is generally unresolved, and this dataset is well suited to investigate this process. Examples of potential buoy surfing events are shown in Mission 18 (Figure 7, panel (a)) and Mission 19 (Figure 8, panel (a)), where all buoys have a sudden change in direction to be nearly directly shoreward within the surf zone. This phenomenon is not observed across all missions. For example, this phenomenon does not occur in Mission 16 (Figure 5, panel (a)). These data can be used to investigate under what conditions this occurs and how it could be further incorporated into models that predict trajectories of buoyant particles. Applications range from scalar transport of plastics to marine search and rescue operations.

Along with the transport of the microSWIFTs, the GPS sensor records the horizontal (east-west and north-south) velocities, and the IMU records the accelerations and rotation rates of the buoy. These data from multiple buoys deployed in a coherent array can be used to investigate the cross and along-shore spatial variability of surface motion. An example of this type of analysis would be comparing the differences in cross-shore velocity and vertical acceleration measured by a buoy inside and outside the surf zone. Figure 11 shows an example of this analysis from Mission 19. In this case, the horizontal velocities are projected into the cross-shore direction, and the vertical acceleration (body frame of reference) is used. These data from all deployed buoys are aggregated and binned into inside and outside the surf zone groups based on the mission's approximate surf zone edge. The cross-shore velocities have been smoothed with a running 1-second mean, and outliers (points greater than 4 standard deviations away from the mean) have been removed. The cut-off location for inside and outside the surf zone was also extended to 1.5 times the approximate surf zone edge. This buffer is added to further separate the types of motion inside and outside the surf zone since intermittent breaking is expected in the outer surf zone, even under the conservative choice of $\gamma_s = 0.35$. In this case, the distribution of horizontal velocities widens and becomes less Gaussian in the tails the distribution inside the surf zone compared to outside, which could indicate the waves are more asymmetric and could also indicate breaking. The distribution of vertical acceleration also becomes less Gaussian inside the surf zone. There is an excess of low acceleration values, consistent with buoys approaching free-fall during active wave breaking (Brown et al., 2019). Future work will extend this analysis to investigate the along-shore variability of these types of surface motion under different wave conditions.

The accelerations and rotation rates measured onboard the microSWIFTs can also be used to investigate the forces and accelerations experienced within breaking waves or bores. Breaking waves manifest as short bursts of high-intensity accelerations (Brown et al., 2019; Sinclair, 2014). We can then identify and describe breaking events as peaks in high acceleration variance. An example of the processing used to locate breaking events is shown for Mission 19 in Figure 12. First, the mean is removed from the vertical acceleration time series from each buoy (single buoy example, Figure 12, panel (a)). The demeaned vertical

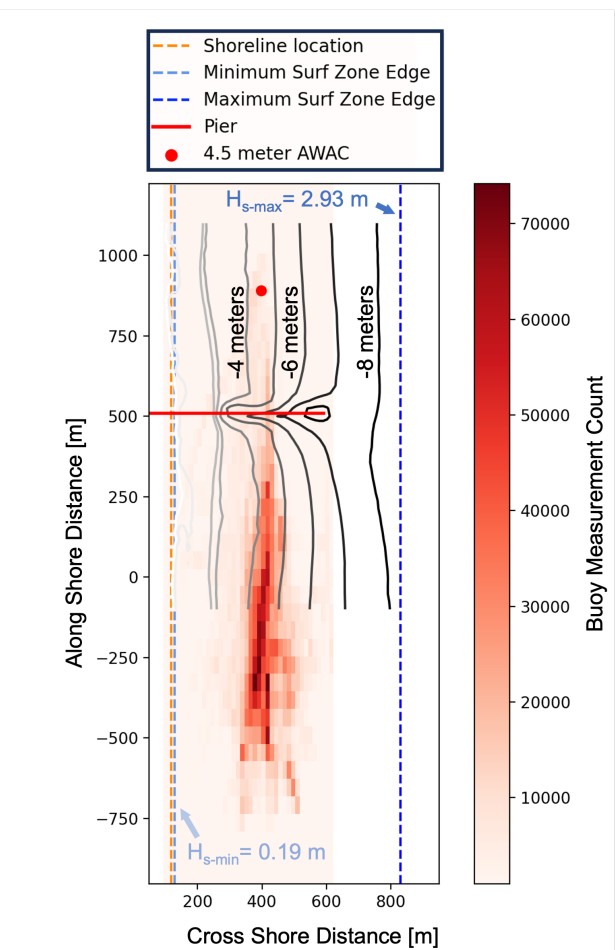

**Figure 10.** Density of Level 1 buoy measurements over the entire experiment from October 3rd to October 30th, 2021, plotted over the bathymetry contours. Most measurements were made on the pier's south side between -2 and -6 meters in bottom elevation. The bin spacing for this histogram is 13.2-meter bins in the cross-shore direction and 54.3-meter bins in the along-shore direction. The average shoreline over the experiment is shown along with the approximate surf zone edges based on the smallest and largest offshore significant wave height during the experiment.

acceleration is then split into five-second (60 data points) windows, and the variance is computed in each window (Figure 12, panel (b)). A threshold of 2.5 times the variance of the entire acceleration time series is used to locate the high-intensity acceleration events or breaking events (Figure 12, panel (b)). This threshold has been tuned empirically; in future analysis, it can be verified with images from the tower at the FRF. Using a threshold of 2.5, we find that the majority of breaking events that the microSWIFTs experience are within the surf zone for Mission 19, as expected (Figure 12, panel (c)).

If this processing method is extended to the entire dataset, we find that most breaking events are within the approximated surf zone with some intermittent breaking outside the surf zone (Figure 13). A total of 3,419 breaking events were detected across

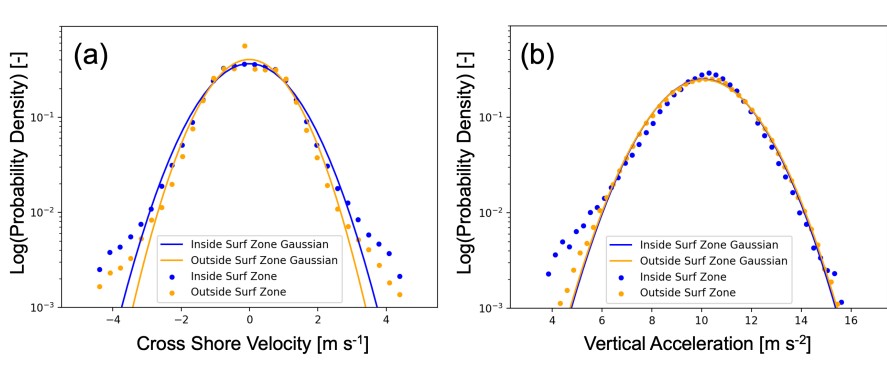

**Figure 11.** Histograms of cross-shore velocity (a) and vertical acceleration (b) from Mission 19. The velocity and acceleration are sorted into inside and outside the surf zone based on the approximate surf zone edge for this mission.

the entire dataset. Using the identified breaking events, we can further investigate the statistics of the motion of the buoys during each breaking event, such as the mean and variance of accelerations and velocities during breaking and the spatial patterns of breaking events under different offshore conditions. These estimates of breaking events can be used to answer questions about the spatial variability of the fraction of breaking waves and energy dissipation in breaking waves.

## 5 Data and Code Availability

The data from the DUNEX experiment are available at https://doi.org/10.5061/dryad.hx3ffbgk0 (Rainville et al., 2023). The dataset consists of netCDF files for each mission, totaling 67 files after all data processing and cleaning. Each netCDF file contains metadata on the mission, including the people who worked on the deployments, deployment style, and Level 1 and Level 2 data, along with all associated metadata. The code to process the Level 0 data to Level 1 and 2 data and to build the final dataset is stored in a GitHub repository at https://github.com/SASlabgroup/DUNEXMainExp. The code used to analyze the data is also contained in the same repository and can be used as an example code to start future analyses.

## 6 Conclusions

We created a unique dataset of measurements of surface kinematics in the nearshore region by using large, coherent arrays of microSWIFT buoys. The Level 1 data consists of measurements of horizontal velocities and positions in the Earth reference frame, three-axis accelerations, rotation rates, and magnetic heading in the buoys' reference frame. The Level 2 data products consist of three-axis accelerations corrected to the Earth reference frame (NED reference frame) by applying a 9-degrees-of-freedom indirect Kalman filter, followed by band-pass filtering. The other Level 2 data products are vertical velocity and sea surface elevation, computed using the corrected vertical acceleration time series. We evaluated individual microSWIFT buoys by comparing spectral energy density and significant wave height estimates with estimates from nearby fixed acoustic

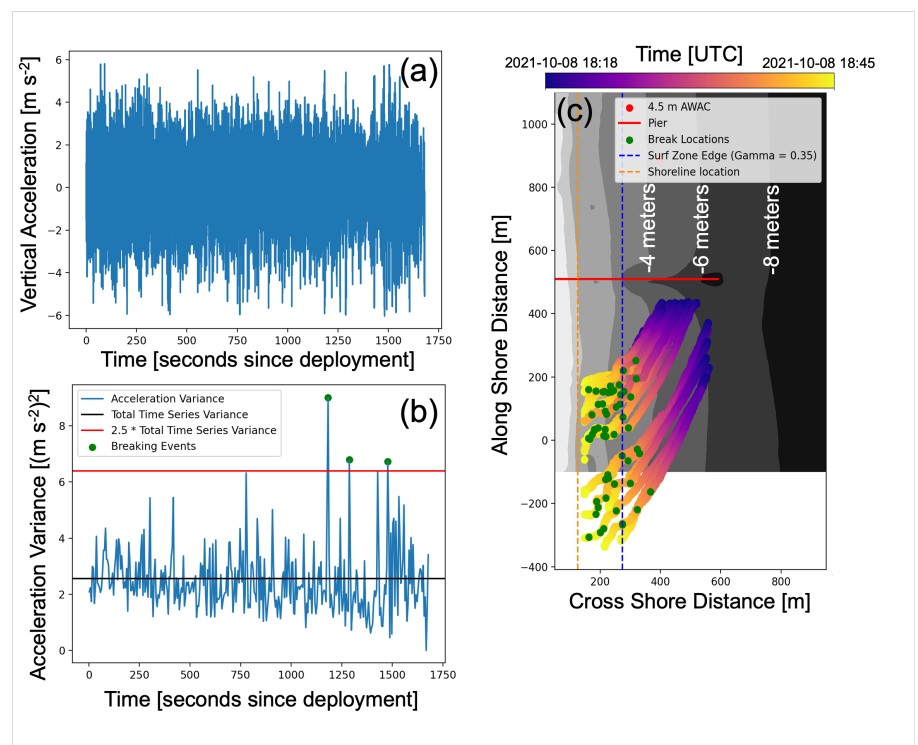

**Figure 12.** Example processing of Mission 19 to locate wave breaking events from high-intensity vertical acceleration peaks. Panel (a) shows an example demeaned vertical acceleration time series. The vertical acceleration time series is split into five-second (60 data point) windows, and variance is computed for each window (b). The variance of the overall vertical acceleration time series and the breaking event threshold (2.5 times the overall vertical acceleration variance) are shown as horizontal lines. The detected breaking events are shown for the example acceleration time series. The spatial location of all detected breaking events is shown along the drift tracks of each microSWIFT for Mission 19 (c).

wave and currents sensors and bottom-mounted pressure gauges. We then used a zero crossing algorithm on the sea surface elevation time series from each microSWIFT to extract individual realizations of measured waves in the field. By aggregating the realizations of waves across each microSWIFT on a mission, we estimate the significant wave height and compare that of the nearby acoustic waves and current sensor estimates. The coherent arrays provide high spatial and temporal resolution measurements during each deployment. Over the experiment, we deployed 81 arrays ranging from 2 to 50 microSWIFTs. Post-
data cleaning left 67 complete missions across the dataset. These 67 missions resulted in a total of 971 drift tracks. 116,307 wave realizations were measured over the experiment, and 3,419 breaking events were detected. These data will be used to investigate nearshore wave kinematics, transport of buoyant particles, and wave breaking processes in a wave-averaged and wave-by-wave framework.

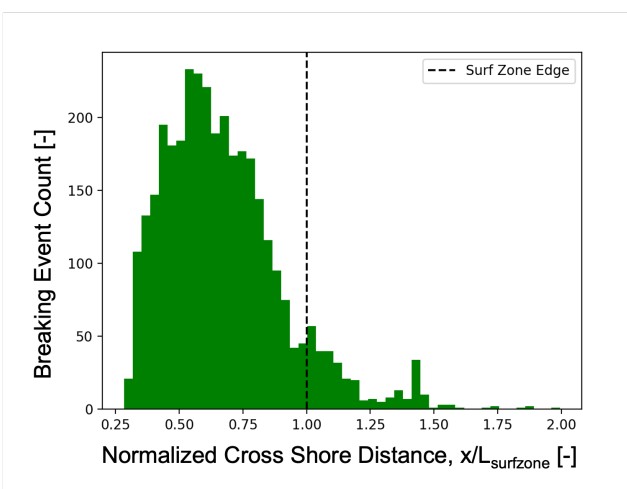

**Figure 13.** Cross-shore variability of detected breaking events across the entire DUNEX experiment. Cross-shore distance is normalized by the approximate surf zone edge of each mission. A total of 3,419 breaking events were detected across the experiment.

*Author contributions.* All authors participated in the data collection, analysis, and writing. Rainville developed the analysis software and prepared the archival dataset. Thomson, Moulton, and Derakhti conceived of the original project and obtained the funding.

*Competing interests.* The authors declare that they have no conflict of interest.

*Acknowledgements.* This work was completed as part of the During Nearshore Event Experiment (DUNEX), which was facilitated by the U.S. Coastal Research Program (USCRP). We thank USCRP for their support of this effort through funding for logistics and coordination (W912HZ-19-2-0045). We would like to thank field engineers Alex de Klerk, Joe Talbert, Emily Iseley, and Nate Clemmet for their efforts in designing, manufacturing, and deploying the microSWIFTs. We also thank Christine Baker, Emma Nuss, Sean McGill, and Jacob Davis for their support in the field. We thank the Town of Duck surf rescue for their help in deploying and retrieving the microSWIFTs throughout the field experiment. We also thank the U.S. Army Engineer Research and Development Center's Field Research Facility for the use of their facility and support staff. We would like to specifically thank Patrick Dickhudt, Mike Forte, Spicer Bak, and many others at the field research facility for all the help they provided in the collection and publication of these data. We thank the reviewers for helping to improve the manuscript and highlight the kinematic measurements.

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
