# Peer review of "Measurements of Nearshore Ocean-Surface Kinematics through Coherent Arrays of Free-Drifting Buoys"

_Earth System Science Data, 2023_

## Referee Comment (RC1)

I recommend publication. SWIFT buoys are used in a range of settings, and comparison with ground truth is valuable.

This paper compares wave statistics from a fixed (AWAC) in 4.5m depth with wave height (H) statistics with statistics from a group (between 10-25) of free floating micro-SWIFT drifters released in about 5m depth.  SWIFT drifted shoreward to the beach where they were recovered. Most  SWIFT observations were between  2-6m depth. The single ground truth in 4.5m depth is inadequate  SWIFT performance for H in the surfzone is not addressed. Wave shape (skewness/asym) is not discussed. The comparison shown (Fig 10b) suggests SWIFT errors night be larger than the errors expected from modern numerical wave models.  Wave shape (e.g. skewness/asymmetry) is not discussed.

Last sentence in abstract: "*These data will be used as a validation dataset for wave-averaged and wave-resolving models and will be used to investigate nearshore wave dynamics. "*  The authors see a robust instrument, "*suitable for investigating dynamics of nearshore waves in both a statistical and wave-by-wave framework."*  I see an instrument not ready for prime wave-time.  Given fig 10, I cannot conjure a nearshore wave dynamics question that could be investigated confidently with SWIFT buoys.  The authors should include a plausible example dynamics investigation.  "Beauty is in the eye of the beholder" certainly  holds here.

235   subset of waves, we compute the significant wave height as the mean of the third-largest waves in the distribution. To calculate a significant wave height from one of these subsets of data, we require at least 30 wave realizations in the distribution. Thus, we do not compute a significant wave height for every mission. We compare the computed significant wave heights to those from the 4.5 m AWAC (Figure 10). Panel (a) shows that the time series of significant wave height from the 4.5 m AWAC and the estimates from the microSWIFT arrays qualitatively agree. Panel (b) directly compares the significant wave heights

240   between the 4.5 m AWAC and the microSWIFT arrays. The linear regression between the 4.5 m AWAC and microSWIFT array significant wave heights has a slope of 1.08 and an $R^2$ value of 0.67, showing a strong correlation between the two significant wave height estimates. This agreement is reasonable given that the microSWIFTs are measuring at a different alongshore location than the AWAC, although in similar water depths. We also expect that the microSWIFT arrays may under-predict some significant wave heights as the sampling windows are shorter than the AWAC, potentially not measuring the largest and

245   least likely waves in the distribution. Nevertheless, the strong agreement in significant wave height and scalar energy density spectra supports that the microSWIFT data are robust and suitable for investigating the dynamics of nearshore waves in both a statistical and wave-by-wave framework.

This 2022 reference is incomplete.
Rainville, E. J., Thomson, J., Moulton, M., and Derakhti, M.: Measurements of Nearshore Waves through Coherent Arrays of Small-Scale, Free-Drifting Wave Buoys, 2022.

Fig 7 : *One error bar is shown for the largest confidence interval of the spectra with 51 degrees of freedom.*
What does "Largest: mean? All confidence limits are the same on a log plot?

[Figure]

**Figure 10.** Comparison of the estimated significant waves heights from the microSWIFT arrays to the estimates from the 4.5 m AWAC. While the microSWIFT arrays are not in the same water depth as the 4.5 m AWAC we see that the microSWIFT array characterizes the size of the waves with good comparison to the 4.5 m AWAC. The gray bars indicate error bars ($\pm$1 standard deviation of the top third largest wave heights) around each of the significant wave height estimates.

---

## Referee Comment (RC3)

**Review of « Measurements of Nearshore waves through Coherent Arrays of Free-Drifting Wave Buoys » Rainville et al., ESSD, 2023**

**General Comments**

This article reports on the development of the microSWIFT, a small wave buoys equipped with a GPS module and an IMU, building on the SWIFT buoys (Thomson, 2012), with the objective of providing measurements to investigate nearshore wave dynamics through the deployment of coherent arrays of microSWIFTs. As a proof of concept, several experiments were conducted at the FRF at Duck, NC, and the paper further presents data processing procedures, the results of which are compared with measurements from a fixed AWAC. The need for more nearshore measurements is unquestionable, and every related research efforts are thus welcome. I appreciate the authors detail the conception of the microSWIFT and how to manage the deployment and some aspects of data processing, although I have some technical concerns regarding the latter (see specific comments below). More generally, I have some doubt regarding the concept of employment, but I guess further research efforts beyond the scope of this study are needed, building on the dataset that have been constituted and made available to the community.

Marc Pezerat, marc.pezerat@shom.fr

**Specific Comments**

1. I.42 "However, it is challenging..." In situ measurements in the nearshore area are indeed challenging, however it is worth mentioning here some recent studies that reported comprehensive field campaign using fixed sensors in such environment (e.g. Guerin et al. 2018, Pezerat et al. 2022, Lavaud et al. 2022)

2. I. 43 "As an alternative..." I rather disagree, the use of Lagrangian device such as wave buoys faces inherent limitations for measuring steep waves that are typically found in the surf zone owing to the simultaneous vertical and horizontal motion of the buoy, the waves in the record tend to look more symmetrical around the mean sea level than they actually are such that non-linear effects cannot be properly investigated with a buoy (e.g. Magnusson et al., 1999; Foristall, 2000). Furthermore, remote sensing techniques, which ability to measure nearshore waves have been demonstrated in a myriad of studies, should be mentioned here.

3. l. 44 "Free drifting buoys..." This assertion should be supported with some appropriate references, furthermore the sentence reads oddly (it looks like a word is missing), I suggest you reword it.

4. l. 51 "... however, they are limited to..." As pointed out above, I would say it is actually an inherent limitations of wave buoys.

5. Fig. 3. The cap on the bottle seems fairly "standard", have you encountered any problems with the seal?

6. I. 99-107 Is there an SD card to keep a record of the data, if not would it be worthy?

7. l. 123-124 "For nearshore applications..." After reading section 2.3, I see nothing in the processing method to account for non-linear effects. As pointed out above this is quite an important limitation of the concept of employment of these buoys, could you thus please elaborate a bit more?

8. I. 136-139 "The microSWFTS were retrieved..." have there been any losses, if so it is worth mentioning the rate of lost bottles such as the reader can have a proper idea of what involve such deployment?

9. I. 153 "Gaps are rare..." Is it also true for GPS data, as it seems to me it is quite common to find some gaps in GPS wave buoys measurements, presumably associated with waves passing over the buoys?

10. I. 201-202 "we use data when an individual microSWIFT..." Why not considering measurements from buoys inside a circle, centered on AWAC location, with a given radius, according to the bathymetry constraint? Here you might have considered data from buoys at quite different locations along the isobaths.

11. l. 204-207 "The spectra are computed..." I get a bit lost here with the estimate of the number of DOF and the resulting spectral resolution. My understanding is that the 10 min (600 sec) records are divided into three overlapping windows (Nw=3) of 300 sec with a 50% overlap, and then, the average spectrum is band-averaged on five frequency bins (M=5). The number of DOF could be thus roughly estimated as: DOF = 2\*Ns\*(M+1)/2 = 18, as opposed to 51. Could you detail a bit more the way spectra are computed?

12. I. 210-211 and I.247-248 I am not convinced by the robustness of the spectral analysis. As pointed out in the two comments above I have some doubts on the way spectra are computed. Furthermore, the spectra show discrepancies that might result in quite important differences on bulk parameters, maybe not Hm0, but what about mean periods? These statements should be tone down, I would rather speak of a relatively good qualitative agreement.

13. l. 215-218 "Since the microSWIFTs..." In practice how individual waves measured by different buoys are tagged? Could you detail a bit more the "sampling with replacement method"?

14. I. 222-226 and Fig. 8e I am not convinced of the relevance of such an aggregated distribution, as the buoys did not measure the same sea state as they drift; what is the meaning of this significant wave height? The following of the paragraph makes more sense to me. I suggest to remove Fig. 8e and the associated discussion.

15. I.233-247 Did you processed AWAC measurements the same way, i.e. zero-crossing processing method using AST measurements or did you consider the spectral estimates of the significant wave height? For sake of clarity, I suggest you dedicate an appendix to the processing of AWAC measurements.

**References:**

Forristall, G. Z. (2000). Wave crest distributions: Observations and second-order theory. *Journal of physical oceanography*, *30*(8), 1931-1943.

Guérin, T., Bertin, X., Coulombier, T., & de Bakker, A. (2018). Impacts of wave-induced circulation in the surf zone on wave setup. *Ocean Modelling*, *123*, 86-97.

Lavaud, L., Bertin, X., Martins, K., Pezerat, M., Coulombier, T., & Dausse, D. (2022). Wave dissipation and mean circulation on a shore platform under storm wave conditions. *Journal of Geophysical Research: Earth Surface*, *127*(3), e2021JF006466.

Magnusson, A. K., Donelan, M. A., & Drennan, W. M. (1999). On estimating extremes in an evolving wave field. *Coastal Engineering*, *36*(2), 147-163.

Pezerat, M., Bertin, X., Martins, K., & Lavaud, L. (2022). Cross-shore distribution of the wave-induced circulation over a dissipative beach under storm wave conditions. *Journal of Geophysical Research: Oceans*, *127*(3), e2021JC018108.

**Technical Corrections**

I. 73 "our team" and elsewhere, please avoid the "we" and the "us".

I. 84 and below the positions should be given with the appropriate distance unit from the origin of the local frame, I assume meters.

I. 93 "mean wave period" are you referring to Tm01 or Tm02 or the mean period issued from waveby-wave analysis? This is in line with my comment 15 above.

---

## Author Comment (AC1)

Responses to Reviewers on *Measurements of Nearshore Waves through Coherent Arrays of Free-Drifting Wave Buoys*

The Reviewer Comments are in red.
The Author's responses are in black.

General Response:

The initial version of this paper focused on estimating wave height statistics from the buoy dataset. Further reflection and the reviewer comments have prompted us to shift our focus to the utility of the Level 1 motion data, including the quality-controlled GPS locations and velocities of the buoys and the body-reference-frame accelerations and rotation rates. These data contain rich information on the kinematics of the ocean surface, how buoyant particles move in the nearshore, and the wave-breaking process. Wave height statistics and other Level 2 data are still a worthwhile avenue to pursue with this dataset; however, we have more clearly stated the caveats and challenges in estimating wave statistics from arrays of buoys rapidly transiting the nearshore region, which are sometimes moving with mean flows and sometimes with breaking waves and bores. In addition to text and figure changes reflecting this shift in focus, we have made a small change to the title: "Measurements of Nearshore Ocean-Surface Kinematics through Coherent Arrays of Free-Drifting Buoys." We have also added a section in the paper to discuss how these data can be used in further studies to address a few of the reviewers' concerns regarding applications. We have also expanded the analysis of the differences in significant wave height between the microSWIFT arrays and the 4.5-meter AWAC. Two other fixed instruments, the 6-meter AWAC and 8-meter array, have been added to the comparison to corroborate the measurements from the 4.5-meter AWAC. More specific comments are addressed below.

Dear Reviewer #1,

Thank you for your thoughtful review. We will address each comment you had in the following response.

Last sentence in abstract: "These data will be used as a validation dataset for wave-averaged and wave-resolving models and will be used to investigate nearshore wave dynamics. " The authors see a robust instrument, "suitable for investigating dynamics of nearshore waves in both a statistical and wave-by-wave framework." I see an instrument not ready for prime wavetime. Given fig 10, I cannot conjure a nearshore wave dynamics question that could be investigated confidently with SWIFT buoys. The authors should include a plausible example dynamics investigation. "Beauty is in the eye of the beholder" certainly holds here.
The first version of this manuscript focuses on the water surface elevation estimates and associated statistics as the key observations provided by the buoys. Taking a step back and considering your review, we reframed the paper to focus on the Level 1 motion data, including horizontal GPS velocities and positions, body frame accelerations, and rotation rates. These are all robust measurements from previously well-established sensors. The Level 2 data products

(i.e., water surface elevation estimates and earth-frame statistics) are still included in the revised paper, with an appropriate description of uncertainties.  We have also modified the paper to give specific examples of how these Level 1 data (i.e., direct motion measurements)  can be used to investigate nearshore dynamics.

The buoys' velocities, accelerations, and rotation rates contain information about how buoyant objects move in the nearshore under varying forcings.  For example, what is the range of accelerations and thus forces that are experienced by a floating object in a nearshore breaking wavefield? How does this change with different offshore conditions? How do the distributions of accelerations and velocities experienced by a buoyant particle change with location in the nearshore?  What is the resulting cross-shore transport?

In another example, the Level 1 motion data provide indicators about the onset and strength of breaking, which may be used to investigate breaking parameterizations.  Here we can extend ideas from previous work, including 1) breaking onset as a critical ratio of particle speed to wave speed (Derakhti et al, 2020), and 2) breaking strength as the instantaneous acceleration at breaker impact (Brown et al, 2018).

We plan to investigate these questions in future papers. Thank you for helping reframe the potential of this dataset.  Please see the *Data Use* section of the revised manuscript for multiple examples of questions that we can answer using this dataset.

The single ground truth in 4.5m depth is inadequate SWIFT performance for H in the surfzone is not addressed.
You are correct in that many assumptions go into comparing the wave heights from microSWIFT arrays with the significant wave height at the 4.5 meter AWAC. To expand the comparison, we now include significant wave heights from the nearby 6 meter AWAC and 8 meter pressure sensor array.  We adjust the measurements from these fixed instruments using linear wave theory to "shoal" the significant wave height measurements to the depth of microSWIFT measurements.

In addition to shoaling, the true wave heights at the microSWIFT array locations might differ from these fixed measurements because of alongshore variations, including refraction and shadowing near the pier.  The fixed instruments are on the north side of the FRF pier, and the microSWIFT arrays were (mostly) on the pier's south side.  The bathymetry around the pier is scoured, and refraction away from this deep feature can reduce the total wave energy arriving at the south side (when waves are arriving from the north).  These effects are now considered in the wave height comparison.

Wave shape (skewness/asym) is not discussed.
The true geometric wave shape, including skewness and asymmetry, is challenging for the buoys to measure accurately. A discussion of these challenges is presented in Lines 67-69: "While buoys have inherent challenges in measuring nearshore waves, including distortion of surface elevation from accelerometer measurements (Magnusson et al., 1999) and inability to

resolve second-order non-linearity (Forristall, 2000), they are the only tool that can be used to obtain direct measurements of the kinematics of the surface."

Though we cannot use the sea surface elevation time series to investigate wave shape directly, we can explore kinematics.  We now include a statistical exploration of the velocities and accelerations for buoys inside and outside the surf zone, and we show non-Gaussian motion within the surf zone. This example is in Figure 11 (shown below), and the discussion surrounding this example analysis is on Lines 330-334 and reads: "In this case, the distribution of horizontal velocities widens and becomes less Gaussian in the tails of the distribution inside the surf zone compared to outside, which could indicate the waves are more asymmetric and could also indicate breaking. The distribution of vertical acceleration also becomes less Gaussian inside the surf zone. There is an excess of low acceleration values, consistent with buoys approaching free-fall during active wave breaking (Brown et al., 2019). Future work will extend this analysis to investigate the along-shore variability of these types of surface motion under different wave conditions."

[Figure]

Figure 11. Histograms of cross-shore velocity (a) and vertical acceleration (b) from Mission 19. The velocity and acceleration are sorted into inside and outside the surf zone based on the approximate surf zone edge for this mission.

The comparison shown (Fig 10b) suggests SWIFT errors might be larger than the errors expected from modern numerical wave models.
The error bars originally shown showed the range of one standard deviation of the wave height distribution. We have since changed these error bars using a bootstrap method to estimate the 95% confidence interval of the significant wave height. This drastically reduces the error bars for missions with sufficient data, and missions with less data now have much larger error bars, as expected. Using the missions with sufficient data will allow us to compare the microSWIFT measurements to models with similar errors.

This 2022 reference is incomplete. Rainville, E. J., Thomson, J., Moulton, M., and Derakhti, M.: Measurements of Nearshore Waves through Coherent Arrays of Small-Scale, Free-Drifting Wave Buoys, 2022.
This citation was corrected.

*Fig 7 : One error bar is shown for the largest confidence interval of the spectra with 51 degrees of freedom. What does "Largest" mean? All confidence limits are the same on a log plot?*

Confidence limits on a log plot are the same size for spectra computed with the same number of degrees-of-freedom (DOF). In Figure 7, each spectrum has a slightly different number DOF, due to different lengths of the time records (all approximately 10 minutes). We show the confidence interval for the spectrum with the lowest number of DOF (=53). The text has been revised to describe how the degrees of freedom are computed for the wave spectra on Lines 236-246. The text discusses the following: "The microSWIFT spectra are computed using Welch's method, with Hanning windows and 50% overlap between adjacent windows. The energy in each five adjacent frequencies is band-averaged to improve the statistical robustness of each estimate. The equivalent degrees of freedom for each spectrum is computed using the formulation in equation 2 for Hanning windows from Thomson and Emery (2014).

$$DOF = (8/3)\frac{N}{M}$$ (Equation 2)

Here, N is the number of data points in the time series, and M is the half-width of the window in the time domain. For these spectra, N = 7200, which is 10 minutes (600 seconds) sampling at 12 Hz frequency, and M = 1800, which is the half-width of a single window. After band-averaging the five adjacent estimates, this results in approximately 53 degrees of freedom (rounded to the closest integer). The AWAC measurements consist of a 34-minute record with a sample rate of 2 Hz, and spectra are 250 computed with 13 50%-overlapping windows (512 points per window) leading to approximately 42 degrees, comparable to that of the microSWIFTs (Christou et al., 2011)."

[Figure]

Figure 7. Comparisons of Panel (a) shows the drift tracks of the microSWIFTs from mission 18 plotted over the surveyed bathymetry DEM. Panel (b) shows a subset of the drift tracks where the bathymetry along each track is between -4.3 and -5.3 meters, and each microSWIFT is a different color. Panel (c) shows the spectra computed from a subset of the sea surface elevation time series for each microSWIFT. One error bar is shown for a confidence interval of the spectra with 53 degrees of freedom. Significant wave heights are computed by numerically integrating the AWAC and averaged microSWIFT spectra.

---

## Author Comment (AC2)

Responses to Reviewers on *Measurements of Nearshore Waves through Coherent Arrays of Free-Drifting Wave Buoys*

The Reviewer Comments are in red.
The Author's responses are in black.

General Response:

The initial version of this paper focused on estimating wave height statistics from the buoy dataset. Further reflection and the reviewer comments have prompted us to shift our focus to the utility of the Level 1 motion data, including the quality-controlled GPS locations and velocities of the buoys and the body-reference-frame accelerations and rotation rates. These data contain rich information on the kinematics of the ocean surface, how buoyant particles move in the nearshore, and the wave-breaking process. Wave height statistics and other Level 2 data are still a worthwhile avenue to pursue with this dataset; however, we have more clearly stated the caveats and challenges in estimating wave statistics from arrays of buoys rapidly transiting the nearshore region, which are sometimes moving with mean flows and sometimes with breaking waves and bores. In addition to text and figure changes reflecting this shift in focus, we have made a small change to the title: "Measurements of Nearshore Ocean-Surface Kinematics through Coherent Arrays of Free-Drifting Buoys." We have also added a section in the paper to discuss how these data can be used in further studies to address a few of the reviewers' concerns regarding applications. We have also expanded the analysis of the differences in significant wave height between the microSWIFT arrays and the 4.5-meter AWAC. Two other fixed instruments, the 6-meter AWAC and 8-meter array, have been added to the comparison to corroborate the measurements from the 4.5-meter AWAC. More specific comments are addressed below.

Dear Reviewer #2,

Thank you for your comments on our work. We will address each of your concerns in the following comments.

This article compares surface gravity wave statistics as measured by GPS and IMU equipped "microSWIFTs", or "mSs" for short (essentially a drifter), to an AWAC (an ADCP). Although this article is informative and should be published, I believe it should only be published after minor to major revision.

I will first comment on a couple "bigger" concerns that I have.

1) The main point of this article is to compare the significant wave height from small drifters equipped with an IMU and GPS to a nearby AWAC. I think the main difficulty with this analysis is that the mS's drift past the depth of the AWAC quite quickly, such that there is relatively few mS observations at the same water depth as the AWAC. The authors are aware of this (eg. line 40) and this is discussed at line 201, 212, 234 etc.

As such only mSs close to the AWAC depth are used in the comparison and there is few mS observations used in the comparison. I believe that the authors might be able to get more mS data to compare as long as the mSs are not within the inner surfzone with active wave breaking. I believe this as Hs should be similar offshore to the break point as it only increases slightly (~10%) as waves shoal to the break point. The critical spot is the break point and not so much the exact same depths. Perhaps the authors could get more data this way?

Thank you for this suggestion. We have adjusted the analysis to include data outside the approximate surf zone for a more complete comparison. This updated comparison between the microSWIFT significant wave height and that from the 4.5 meter AWAC is shown in Figure 9 (see below). Including more data from the measurements outside the surf zone has improved the comparison. In this figure, we also include other fixed sensors, corrected for shoaling, to confirm the measurements from 4.5 meter AWAC. We have also added analysis as to whether the microSWIFT array is within the shadow of the pier, which can lead to an underestimation of the significant wave height. This analysis helps to explain many of the cases where the microSWIFT array underestimates the significant wave height but does not provide an explanation for all points. Other underestimated points could be explained as being far from the sensor and thus seeing different waves than the fixed sensors.

If they can, I would like a more detailed comparison between the spectra of the mSs and AWAC and not just the mSs that have similar depths as the AWAC. Currently the only comparison in the MS is between the AWAC and mS Hs and a_0(f), the SSH spectra. This is a limited comparison. I think the analysis should be extended to compared mean periods, mean directions, and directional spreads. The analysis needs to be expanded to include aspects of the wave field in addition to the significant wave height.

We appreciate this suggestion; however, we have shifted our perspective on the utility of the dataset given the comments of all the reviewers and decided that it is more appropriate to pursue the Level 1 data products further. Therefore, we have kept just a single example comparison of the energy density spectra from the microSWIFTs to the 4.5 meter AWAC, shown in Figure 7. Future studies may look more into other wave statistics that can be computed with the microSWIFTs, including mean periods, mean direction, and directional spread. While we present an example comparison of the energy density spectra to the fixed instruments, we acknowledge that this comparison is not direct since the mircoSWIFTs are not in the same location as the AWAC. Further studies may use a more dedicated verification process for the Level 2 data that could be applied to this dataset. For now, we change the focus of the paper to look at the dataset's ability to investigate surface transport, spatial variability of surface kinematics, and detection of breaking waves and the associated dynamics of the breaking waves. Please see the *Data Use* section of the revised manuscript for a description of how this dataset can be utilized.

I also believe that the authors should calculate Hs as the integral of the spectra over sea-swell frequencies and not from the distribution of wave heights as they should be the same. If possible Hs from AWAC and mSs should be calculated in the exact same manner.

For the example spectra comparison, the significant wave height has been computed from the integral of the spectral energy density in this example, as shown in Figure 7 below. This

example comparison has a good qualitative agreement between the calculated significant wave heights, shown in the legend of the figure. However, since the buoys are drifting quickly through the surf zone, there is rarely enough data to compute an energy density spectrum from a single buoy. Instead, we rely on the zero crossing method and aggregating wave realizations (i.e., individual wave heights) across multiple buoys to estimate the significant wave height. This approach also highlights our intention for "coherent" array analysis.

[Figure]

Figure 7. Comparisons of Panel (a) shows the drift tracks of the microSWIFTs from mission 18 plotted over the surveyed bathymetry DEM. Panel (b) shows a subset of the drift tracks where the bathymetry along each track is between -4.3 and -5.3 meters, and each microSWIFT is a different color. Panel (c) shows the spectra computed from a subset of the sea surface elevation time series for each microSWIFT. One error bar is shown for a confidence interval of the spectra with 53 degrees of freedom. Significant wave heights are computed by numerically integrating the AWAC and averaged microSWIFT spectra.

Also, getting the significant wave height as the mean of the top 1/3 wave heights (line 235) isn't as straight forward as

$Hs = <H^2>^{1/2}$

or

$Hs = (8/pi)^{(1/2)} <H>$   (1.6 times the mean of all wave heights)

where H is the random wave heights. Recall, $Hs = 4<eta^2>^{1/2}$ and eta=H/2. As, the distribution of H is Rayleigh distributed, there is only 1 free parameter (Hs). see

https://en.wikipedia.org/wiki/Rayleigh_distribution

it would be more straight forward to calculate Hs this way and it should be more robust than the mean of the top 1/3 waves (which limit the number of waves you use by 1/3!). The mean of top 1/3 merely a consequence of a Rayleigh distribution with only 1 free parameter (and gets rid of the 1.6 factor for all waves). And it would be easier to derive error bars (see link above).

Thank you for suggesting this change; we have revised our approach based on this idea.  We now compute the root-mean-square wave height, which uses all measured wave heights and is less sensitive to outliers. The significant wave height is then computed from the RMS wave height as described in Lines 272-273 as follows: "The significant wave height is computed by first computing the root-mean-square of the wave heights and then multiplying by a factor of 1.416 to convert to significant wave height for a Rayleigh distribution (Dean and Dalrymple, 1991)."

2) This article purports to compare wave statistics from mS's and an AWAC. And the article does compare Hs (significant wave height) between the two instruments. The article also compares the SSH (see surface height) frequency spectra between the instruments. I believe this article would be greatly improved if additional statistics  were compared such as mean direction and directional spread (like in Raghukumar et al 2019). See also comment above.

Thank you for this comment. Since receiving comments from all the reviewers, we have shifted our perspective on the best use for the dataset and decided that focusing on the Level 1 data is the best use for this dataset. The analysis and comparison to other instruments completed by Raghukumar et al. 2019 is excellent. After our shift in perspective to focus on the Level 1 data, we think that this is a more appropriate use of the data that also differentiates the microSWIFTs from the Spotter buoys. See the general comments above for more detail, and see the *Data Use* section of the revised manuscript. For a full comparison of dirential moments with a Spotter buoy please see *Development and testing of microSWIFT expendable wave buoys* (submitted draft to Coastal Engineering Journal, submitted version is available here: https://github.com/SASlabgroup/SWIFT-codes/blob/master/Documents/microSWIFTs_CEJsubmission_9Jul2023.pdf).

3) I believe that a more detailed exploration of the uncertainty in Hs (for instance) should be explored. What is the 95% CL of the AWAC Hs? What is the 95% CL on the mS estimate. The authors should look at Gemmrich et al 2016 regarding this calculation. Are the differences in Hs between mS and AWAC real or statistical? The article would be greatly improved if the authors can state whether the differences in Hs between the instruments are within expectations. If there are real differences in Hs, reasons for the differences should be explained.

We have now included a more detailed exploration of the uncertainty in the estimates of significant wave height, shown in Figure 9 (see below). We computed 95% confidence intervals using a bootstrap method. We see that some points, especially those that overestimate the significant wave height, have very wide confidence intervals indicating that these points are computed with few data points. We have also included an analysis of when the microSWIFT arrays are within the 'shadow' of the pier, which previous authors have shown reduces wave energy significantly. This analysis helps to explain why some points underestimate the significant wave height. We have also added analysis to show when arrays are within a 300 meter radius of the 4.5 meter AWAC, and we see that the remaining points that underestimate the significant wave height, not explained by being within the shadow of the pier, are further away from the AWAC which could also indicate spatial differences in the waves. We now see this analysis as less of a direct comparison between wave heights since they are not measuring the exact location and the estimates are from an aggregate of buoys in different locations.

At Lines 287-296 of the revised manuscript we now state: "The linear regression between the 4.5 m AWAC and microSWIFT array significant wave heights has a slope of 0.61 and an R2 value of 0.74, showing a positive correlation between the two significant wave height estimates. This agreement is reasonable given that the microSWIFTs are measuring at a different alongshore location than the AWAC, although in similar water depths. We also expect that the microSWIFT arrays may under-predict some significant wave heights as the sampling windows are shorter than the AWAC, potentially not measuring the largest and least likely waves in the distribution and times that the microSWIFTs are within the 'shadow' of the pier. Being in the pier 'shadow' is defined here as missions when the average location of the microSWIFTs during a mission is within 200 meters of the pier, and waves are coming from the other side of the pier based on the mean wave direction from the 8-meter array (furthest offshore sensor). The significant wave height measurements from the 6-meter AWAC and 8-meter array are also adjusted to be in the same depths using linear wave theory and compared to show agreement between multiple sensors for a more robust comparison."

4) Only wave statistics are compared in this article but I think the currents could be compared as well. How do mean currents compare? How much of the mean on-off shore velocity can be attributed to Stokes drift? Do the mS's surf broken wave bores?

This is an excellent suggestion and a component we will focus on in future studies. Since the microSWIFT measurements are relatively far away from the fixed instruments, it may not be reasonable to compare the mean currents between the microSWIFTs and the fixed instruments. Further studies may use tracking of foam in video to estimate currents closer to the microSWIFT region. Remote sensing also will be used to track wave crests. These measurements, along with the microSWIFT tracks, can then assess whether the microSWIFT movement is best described

by mean currents, Stokes drift, and/or surfing on broken wave bores. The microSWIFTs were visually observed to surf on the broken wave bores. We are interested in investigating this further, but it is outside of the scope of this data paper. Lines 306-318 now note that: "These measurements can investigate buoyant particles' cross and along-shore transport under various forcing conditions. The microSWIFTs move with both the waves and the currents. They are buoyant, and thus they also tend to 'surf' on the broken waves. The buoys' surfing can enhance the transport of these objects (Pizzo, 2017). This type of motion affects similar objects transported in the surf zone, such as large algae, e.g., Sargassum, a buoyant seaweed affecting coastlines in the south-eastern US (Webster and Linton, 2013). The mean surface currents, Stokes drift, and rip currents are resolved within transport models for surface-constrained particle motion (Moulton et al., 2023). The process of 'surfing' is generally unresolved, and this dataset is well suited to investigate this process. Examples of potential buoy surfing events are shown in Mission 18 (Figure 7, panel (a)) and Mission 19 (Figure 8, panel (a)), where all buoys have a sudden change in direction to be nearly directly shoreward within the surf zone. This phenomenon is not observed across all missions. For example, this phenomenon does not occur in Mission 16 (Figure 5, panel (a)). These data can be used to investigate under what conditions this occurs and how it could be further incorporated into models that predict trajectories of buoyant particles. Applications range from scalar transport of plastics to marine search and rescue operations."

Line by line comments:

Line 41. When discussing wave statistics, the authors state "Fixed sensors generally have robust statistics since they measure continuously for long periods." This feels misleading to me. Most fixed sensors, such as a wave buoy (or ADCP), measures the variance of SSH over a fixed duration, say 30 min, ie. var(eta). Then Hs = 4 * (var(eta))^(1/2). As this is done every 30 min (or 20 min), there is now a time series of Hs. The robustness of a single estimate of Hs doesn't have anything to do with the instrument being fixed as you could get the same estimate from a drifting instrument (see Herbers et al 2012) as long as it sampled for 30 min straight and it samples a statistically stationary wave field. In this article, the issue is that the mS's don't sample a statistically stationary wave field as they drift through the surfzone (see big comment #1 above).

This is a correct interpretation of the differences between the fixed and drifting sensors. There should not be any difference between the two sensors as long as they sample in the same place for at least 30 minutes. However, free drifting platforms do not stay stationary in the surf zone, and therefore never record 30 minutes continuously in the same place as fixed instruments do. In that case, since the fixed instruments sample longer in the same location, they should have more robust statistics than the drifting platform. This is one of the many challenges with the drifting platform.

Line 131. The authors say, "However, when deployed in large numbers as coherent arrays, the mS's can be processed together to explore the spatial variability of the nearshore waves and currents." Is this done here? This article does not explore the spatial variation of the wave field.

We have added some analysis to explore the spatial variability of the surface kinematics. Figure 11 in the revised manuscript now includes an example comparison of cross shore velocity and vertical acceleration inside and outside the surf zone (see Figure 11 below). In this analysis, we look for changes in cross-shore velocities and vertical accelerations, which can both be used to infer properties about the wave field. This analysis uses only Level 1 data. A discussion has been added to the revised manuscript on Lines 320-334: "These data from multiple buoys deployed in a coherent array can be used to investigate the cross and along-shore spatial variability of surface motion. An example of this type of analysis would be comparing the differences in cross-shore velocity and vertical acceleration measured by a buoy inside and outside the surf zone. Figure 11 shows an example of this analysis from Mission 19. In this case, the horizontal velocities are projected into the cross-shore direction, and the vertical acceleration (body frame of reference) is used. These data from all deployed buoys are aggregated and binned into inside and outside the surf zone groups based on the mission's approximate surf zone edge. The cross-shore velocities have been smoothed with a running 1-second mean, and outliers (points greater than 4 standard deviations away from the mean) have been removed. The cut-off location for inside and outside the surf zone was also extended to 1.5 times the approximate surf zone edge. This buffer is added to further separate the types of motion inside and outside the surf zone since intermittent breaking is expected in the outer surf zone, even under the conservative choice of γs = 0.35. In this case, the distribution of horizontal velocities widens and becomes less Gaussian in the distribution's tails inside the surf zone compared to outside, which could indicate the waves are more asymmetric and could also indicate breaking. The distribution of vertical acceleration also becomes less Gaussian inside the surf zone. There is an excess of low acceleration values, consistent with buoys approaching free-fall during active wave breaking (Brown et al., 2019). Future work will extend this analysis to investigate the along-shore variability of these types of surface motion under different wave conditions."

[Figure]

Figure 11. Histograms of cross-shore velocity (a) and vertical acceleration (b) from Mission 19. The velocity and acceleration are sorted into inside and outside the surf zone based on the approximate surf zone edge for this mission.

I believe that it should and show that Hs decreases shoreward consistent with expectations. Inside the surf zone and in the outer surf zone, the microSWIFTs are exposed to breaking waves which manifest as high-intensity accelerations. We take as many precautions as we can to reduce these events in the acceleration signals through digital filtering described in Lines 213-217: "We then use a first-order Butterworth band-pass filter to remove low (f < 0.05 Hz) and high (f > 0.5 Hz) frequency noise outside of the gravity wave band from the signals. We then integrate the filtered acceleration signals via a time domain cumulative trapezoid method to velocities. The velocities are filtered again with the same filter to eliminate any spurious integration errors, then integrated to estimate positions, and finally filtered one last time to eliminate integration errors. The corrected and filtered accelerations, velocities, and positions are the Level 2 data." However, within the surf zone where breaking is prevalent and the wave height is expected to decrease, the buoys have the most high-intensity acceleration events, which leads to spurious large waves. Therefore, within the surf zone, we do not see a reduction in Hs, and we restrict our analysis to outside the surf zone where these events are more intermittent, following your previous suggestion. This is further discussed in the revised manuscript on Lines 223-227: "We also expect measurements outside the surf zone to be more reliable for estimating wave properties since they are exposed to fewer breaking waves. Breaking waves tend to manifest as short bursts of high frequency and amplitude accelerations (Sinclair, 2014; Brown et al., 2019; Feddersen et al., 2023). Integrating these acceleration bursts can lead to spuriously large or nonphysical sea surface elevations; therefore, we expect the best agreement of wave measurements when the buoys are outside the surf zone or in the outer surf zone where breaking is more intermittent."

Line 192. When discussing the AWAC wave statistics, the authors should describe the statistics in much better detail. I suppose that the AWAC gives: $a_0(f)$, the SSH spectra, $a_1(f)$, $a_2(f)$, $b_1(f)$, $b_2(f)$. From these the AWAC estimates: Theta(f), the mean direction at each frequency; and sigma_Theta(f), the directional spread at each frequency. These statistics should be listed in a methods section. In that section, how the AWAC computes these statistics should be stated. ie., How long is the record for which $a_0(f)$ is calculated? The number of degrees of freedom is stated at 48, where does this number come from? Also, the authors state at line 206 that there are 51 degrees of freedom for the mS $a_0(f)$. How does one get an odd number of degrees of freedom for a spectra? I calculate the DOF as 2x3x5=30: 2 for each periodogram, 3 for 3 separate 5 minute chunks of data (these are not completely independent, and 5 for averaging 5 independent frequencies.

The AWAC is a fixed instrument the FRF maintains, and its staff completes all data processing. Since we are not computing the other statistics from the microSWIFT arrays, it may not be helpful to show these statistics. A comparison of directional moments between the microSWIFTs and a Spotter buoy is shown in full detail in *Development and testing of microSWIFT expendable wave buoys* (submitted draft to Coastal Engineering Journal, submitted version is available here: https://github.com/SASlabgroup/SWIFT-codes/blob/master/Documents/microSWIFTs_CEJsubmission_9Jul2023.pdf).

A more detailed explanation of how the degrees of freedom for the microSWIFT and AWAC spectra are computed is described in Lines 241-251. The discussion follows: "The microSWIFT spectra are computed using Welch's method, with Hanning windows and 50% overlap between adjacent windows. The energy in each five adjacent frequencies is band-averaged to improve the statistical robustness of each estimate. The equivalent degrees of freedom for each spectrum is computed using the formulation in equation 2 for Hanning windows from Thomson and Emery (2014).

$$DOF = (8/3)\frac{N}{M}$$

(Equation 2)

Here, N is the number of data points in the time series, and M is the half-width of the window in the time domain. For these spectra, N = 7200, which is 10 minutes (600 seconds) sampling at 12 Hz frequency, and M = 1800, which is the half-width of a single window. After band-averaging the five adjacent estimates, this results in approximately 53 degrees of freedom(rounded to the closest integer). The AWAC measurements consist of a 34-minute record with a sample rate of 2 Hz, and spectra are computed with 13 50%-overlapping windows (512 points per window) leading to approximately 42 degrees, comparable to that of the microSWIFTs (Christou et al., 2011)."

Line 216. "... some of the mS's will be measuring the same wave as it propagates ... which improves the robustness of the statistics by sampling many realizations." I don't think this is correct. In order to increase the robustness (ie. get an estimate closer to the true value), it is necessary to sample different regions (or times) of the wave field (eg Gemmrich et al 2016). If two mS's have the exact same 10 min z(t), the 2 estimates of Hs are the same and I would argue less robust. In this case, having mS's far from each other, so they are not coherent, would increase the robustness of the estimate. Fig 9 in Raghukumar et al 2019 suggests that 50 m is the length scale over which waves decorrelate. This suggests a thorough investigation of the errors in estimating Hs.

Thank you for correcting this phrasing. While measuring multiple realizations does not necessarily improve the robustness of the estimate, it can improve the confidence in the estimate. Further discussion on this idea is given in Lines 262-268, and the discussion is the following: "Since the microSWIFTs are spatially distributed in the nearshore and sampling simultaneously, some microSWIFTs will measure the same wave as it propagates past multiple buoys. We treat this like a physical 'sampling with replacement' method similar to Monte Carlo or Bootstrap simulation methods known as re-sampling techniques. These types of re-sampling techniques are used to improve the confidence in a statistical estimate from a finite amount of data (Thomson and Emery, 2014). In this case, the finite data is the short period that the buoys sample an area, but multiple datasets from different microSWIFTs, occasionally containing measurements of the same wave, can help improve confidence in the statistics."

Fig 1. Add LLW and HHW levels to Fig 1c for reference.
These changes have been added to Figure 1, shown below.

[Figure]

Figure 1. (a) Aerial imagery of the Outer Banks of North Carolina, US, where the gold star is the location of the US Army Corps of Engineers - Field Research Facility (FRF) (© Source: Esri, Maxar, Earthstar Geographics, and the GIS User Community). Panel (b) shows the bathymetry contours at the field site from October 21st, 2021, relative to the NAVD88 datum and locations of fixed instrumentation (Data provided by USACE, Field Research Facility, https://frfdataportal.erdc.dren.mil/). Panel (c) shows the average cross-shore profile of the bathymetry with one standard deviation above and below the average. The higher high water (HHW) and lower low water(LLW) levels measured during the experiment are also shown.

A dashed line for the shoreline location based on an average bathymetry profile and the water level from the nearby NOAA water level gauge. A blue dashed line has also been added to show the location where we expect the surf zone edge to be based on the ratio of offshore significant wave height (from the FRF's 8-meter array of pressure sensors) to water depth. A discussion of how the surf zone edge is estimated is added on Lines 153-167, and the discussion is the following: "For each mission, the mean water level during the deployment is added to the alongshore bathymetry profile to give a cross shore depth profile during the mission. The shoreline is then estimated as the cross shore location where the depth during the mission equals zero on the along shore averaged profile. Waves are expected to begin breaking when the ratio of wave height Hs to water depth d,

$$\gamma s = H_s/d \, , \quad (1)$$

reaches a certain threshold. Using this definition of $\gamma s$, the variable Hs represents the offshore significant wave height (will use measurements from the 8-meter pressure gauge array, location in Figure 1, panel (c)), and the variable d represents the water depth during the mission. Values of $\gamma s$ from the Duck, NC field site have been observed to be between 0.4 and 0.8 (Sallenger Jr and Holman, 1985). Further studies have shown that within the inner surf zone at the Duck, NC field site $\gamma s$ can reach as low as approximately 0.275 and as high as 0.375 at depths greater than 0.8 meters (Raubenheimer et al., 1996). Smaller values of $\gamma s$ drive the breaking depth to deeper water, and larger values drive the breaking depth to shallower waters. From these observed values, we chose $\gamma s = 0.35$ to provide a representative estimate of the surf zone edge location. These estimates are shown in both panels of Figure 5, and the same estimation method is used to add context to the analysis later on. The choice of $\gamma s = 0.35$ is a traditionally low value but is used as a conservative estimate to include the outer surf zone where intermittent breaking is prevalent."

The changes to the figure are shown in Figure 5 below. Panel (a) has measurements both inside and outside of the breakers, while Panel (b) just has measurements inside where we expect breaking to be occurring.

[Figure]

Figure 5. Example drift tracks(location of microSWIFTs over time) of microSWIFT arrays during a mission plotted over of the bathymetry digital elevation model shown in Figure 1 Panel (b). Panel (a) shows the drift tracks from mission 16, which has 19 microSWIFTs deployed, and Panel (b) shows the drift tracks from mission 79, which has 13 microSWIFTs deployed. Approximate shoreline and surf zone edges are shown for each mission.

Fig 7a,b. Add a dashed line where wave breaking starts. Do the mSs sample a region of broken waves? Where is the shoreline. How wide is the surfzone?
Fig 7c. The lines are very hard to tell apart. Choose better colors, especially for the AWAC (thick black?). Also choose some different thicknesses. Hs for each should also be stated in the caption. Maybe also show the mean of all mSs? Then the individual mSs could be thin gray.
This figure has been adjusted with your suggestions. Panels (a) and (b) now include a dashed line for the shoreline and the location we expect breaking to occur, as shown in Figure 7 above. Panel (c) of Figure 7 (shown above) now shows each individual microSWIFT as a thin gray line and the mean value as a dark black line. The significant wave heights computed from integrating the two spectra over the frequency domain are shown in the legend.

Fig 8a,b. Add a dashed line where wave breaking starts. Do the mSs sample a region of broken waves? Where is the shoreline. How wide is the surfzone?
Dashed lines for the shoreline and the surf zone edge have been added to both panels (a) and (b) of Figure 8, as shown below.

[Figure]

Figure 8. Example of steps in processing each mission. Panel (a) shows the drift tracks of the microSWIFTs from mission 19 plotted over the surveyed bathymetry DEM. Panel (b) shows the same drift tracks as Panel (a) but shows each microSWIFT as a different color. Panel (c) show the time series of computed sea surface elevation, with one-time series being highlighted as an example. Panel (d) is a zoomed-in portion of the overall time series showing the locations of zero crossings and how we define the height of an individual wave in a time series. Panel (e) is the probability density of all wave heights from the entire time series, where the colors show the contribution from each microSWIFT with the corresponding color. The probability density distribution fits a Rayleigh distribution. The vertical line shows the computed significant wave height for this distribution and the 95% confidence interval of the estimate.

Fig 9. Not sure if this figure needs to be included. It could be improved too. Use contour rather than contourf for the the bathy.
We have adjusted the original Figure 9, which showed the location of wave realizations, to now show the spatial density of Level 1 data from the microSWIFTs. This shows the spatial range of measurements and where those measurements are concentrated. We primarily sampled on the pier's south side due to logistic restraints but did sample some of the north side of the pier,

especially in the inner surf zone. The figure has been improved with your suggestions, as shown in Figure 10 below (it has been changed to Figure 10 in the revised manuscript).

[Figure]

Figure 10. Density of Level 1 buoy measurements over the entire experiment from October 3rd to October 30th, 2021, plotted over the bathymetry contours. Most measurements were made on the pier's south side between -2 and -6 meters in bottom elevation. The bin spacing for this histogram is 13.2-meter bins in the cross-shore direction and 54.3-meter bins in the along-shore direction. The average shoreline over the experiment is shown along with the approximate surf zone edges based on the smallest and largest offshore significant wave height during the experiment.

Fig 10a,b. std bars are on mSs Hs in 10a but on AWAC Hs in 10b. Shouldn't the error bars in 10b be on mSs as they are the same as in 10a? In 10b, error bars on should be on the AWAC and mS estimates. Also, these should be 95% confidence limits, not stds as the reader will be interested in how good your estimate is and not how much scatter there is in the estimate. Please calculate correct 95% confidence limits based on the number of wave heights used.
Fig 10b. Although overall the scatter is OK (R^2 = .67), for AWAC Hs>2, there is little to no relationship between AWAC and mS Hs. This should be commented on in the Ms. Perhaps including all mSs that are not in the surfzone will make the relationship better? Perhaps, correct 95% confidence limits will help to explain when the relationship isn't so great?
We have implemented your suggestions for the original Figure 10, shown below from the revised manuscript, now as Figure 9. The axes have been flipped to show the 4.5 meter AWAC significant wave height on the horizontal axis and the other significant wave height estimates on the vertical axis. The error bars have been corrected to be 95% confidence intervals estimated using a bootstrap method. We have also removed the original Figure 10a, which did not add information beyond the current Figure 9. Measurements from the 6 meter AWAC and 8 meter array of pressure sensors have been corrected for shoaling and added to this analysis to show how other instruments compare with the measurements from the 4.5 meter AWAC. The microSWIFT array estimates of significant wave height also now use all data outside of the surf zone, which includes many more measurements.

Further analysis to determine when the microSWIFT arrays are within the shadow of the pier has been added and to show when the microSWIFTs are within a radius of 300 meters of the 4.5 meter AWAC. These additional analyses help to explain the discrepancies between the 4.5 meter AWAC measurements and those from the microSWIFTs. Discussion of these additional analyses is included in the revised manuscript on Lines 287-297 and is the following: "The linear regression between the 4.5 m AWAC and microSWIFT array significant wave heights has a slope of 0.61 and an R2 value of 0.74, showing a positive correlation between the two significant wave height estimates. This agreement is reasonable given that the microSWIFTs are measuring at a different alongshore location than the AWAC, although in similar water depths. We also expect that the microSWIFT arrays may under-predict some significant wave heights as the sampling windows are shorter than the AWAC, potentially not measuring the largest and least likely waves in the distribution and times that the microSWIFTs are within the 'shadow' of the pier. Being in the pier 'shadow' is defined here as missions when the average location of the microSWIFTs during a mission is within 200 meters of the pier, and waves are coming from the other side of the pier based on the mean wave direction from the 8-meter array (furthest offshore sensor). The significant wave height measurements from the 6-meter AWAC and 8-meter array are also adjusted to be in the same depths using linear wave theory and compared to show agreement between multiple sensors for a more robust comparison. The agreement in significant wave height and scalar energy density spectra supports that the Level 2 data are useful for investigating wave spectra and statistics."

[Figure]

Figure 9. Comparison of the estimated significant wave heights from the microSWIFT arrays, 6-meter AWAC, and 8-meter pressure sensor array (6-meter AWAC and 8-meter array have been corrected for shoaling) to the estimates from the 4.5 m AWAC. While the microSWIFT arrays are not in the same water depth as the 4.5 m AWAC, we see that the microSWIFT arrays characterize the size of the waves with good comparison to the 4.5 m AWAC. The gray bars indicate 95% confidence intervals around each of the significant wave height estimates, computed using a bootstrap method from the distributions of wave heights. The colors of the estimates depict if the microSWIFT array is in the 'shadow' of the pier where we expect a reduction in wave energy.

References:

1. Dean, R. G. and Dalrymple, R. A.: Water wave mechanics for engineers and scientists, vol. 2, world scientific publishing company, 1991.
2. Thomson, J., Bush, P., Contreras, V., Clemett, N., Davis, J., de Klerk, A., Iseley, E., Rainville, E., Salmi, B., Talbert, J.: Development and testing of microSWIFT expendable buoys. Manuscript submitted to Coastal Engineering Journal. https://github.com/SASlabgroup/SWIFT-codes/blob/master/Documents/microSWIFTs_CEJsubmission_9Jul2023.pdf, 2023.
3. Thomson, R. E. and Emery, W. J.: Data analysis methods in physical oceanography, Newnes, 2014.

4.  Raghukumar, K., Chang, G., Spada, F., Jones, C., Janssen, T., and Gans, A.: Performance Characteristics of "Spotter," a Newly Developed Real-Time Wave Measurement Buoy, Journal of Atmospheric and Oceanic Technology, 36, 1127–1141, https://doi.org/10.1175/JTECH-D-18-0151.1, 2019.
5.  Zinsser, William. "On writing well: The classic guide to writing nonfiction." New York, NY (2006).

---

## Author Comment (AC3)

Responses to Reviewers on *Measurements of Nearshore Waves through Coherent Arrays of Free-Drifting Wave Buoys*

The Reviewer Comments are in red.
The Author's responses are in black.

General Response:

The initial version of this paper focused on estimating wave height statistics from the buoy dataset. Further reflection and the reviewer comments have prompted us to shift our focus to the utility of the Level 1 motion data, including the quality-controlled GPS locations and velocities of the buoys and the body-reference-frame accelerations and rotation rates. These data contain rich information on the kinematics of the ocean surface, how buoyant particles move in the nearshore, and the wave-breaking process. Wave height statistics and other Level 2 data are still a worthwhile avenue to pursue with this dataset; however, we have more clearly stated the caveats and challenges in estimating wave statistics from arrays of buoys rapidly transiting the nearshore region, which are sometimes moving with mean flows and sometimes with breaking waves and bores. In addition to text and figure changes reflecting this shift in focus, we have made a small change to the title: "Measurements of Nearshore Ocean-Surface Kinematics through Coherent Arrays of Free-Drifting Buoys." We have also added a section in the paper to discuss how these data can be used in further studies to address a few of the reviewers' concerns regarding applications. We have also expanded the analysis of the differences in significant wave height between the microSWIFT arrays and the 4.5-meter AWAC. Two other fixed instruments, the 6-meter AWAC and 8-meter array, have been added to the comparison to corroborate the measurements from the 4.5-meter AWAC. More specific comments are addressed below.

Dear Dr. Pezerat,

Thank you for taking the time to review this manuscript. We have changed our perspective on how to use this dataset best and therefore, we have shifted the focus of this study towards the Level 1 data capabilities to present the utility of the dataset. Please see the general comments on the paper above; your specific comments are addressed below.

This article reports on the development of the microSWIFT, a small wave buoys equipped with a GPS module and an IMU, building on the SWIFT buoys (Thomson, 2012), with the objective of providing measurements to investigate nearshore wave dynamics through the deployment of coherent arrays of microSWIFTs. As a proof of concept, several experiments were conducted at the FRF at Duck, NC, and the paper further presents data processing procedures, the results of which are compared with measurements from a fixed AWAC. The need for more nearshore measurements is unquestionable, and every related research efforts are thus welcome. I appreciate the authors detail the conception of the microSWIFT and how to manage the deployment and some aspects of data processing, although I have some technical concerns regarding the latter (see specific comments below). More generally, I have some doubt

regarding the concept of employment, but I guess further research efforts beyond the scope of this study are needed, building on the dataset that have been constituted and made available to the community.

Specific Comments:
1. l.42 "However, it is challenging…" In situ measurements in the nearshore area are indeed challenging, however it is worth mentioning here some recent studies that reported comprehensive field campaign using fixed sensors in such environment (e.g. Guerin et al. 2018, Pezerat et al. 2022, Lavaud et al. 2022)

Discussion about successful, comprehensive field campaigns that have used fixed sensors has been added in Lines 46-49 and includes your suggested citations. The section where this is addressed is now the following: "While it can be challenging to deploy many fixed sensors and remote sensing systems, many field campaigns have been successful using these methods in the nearshore region (Guérin et al., 2018; Pezerat et al., 2022; Lavaud et al., 2022; Carini et al., 2015; Elgar et al., 2001)."

2. l. 43 "As an alternative…" I rather disagree, the use of Lagrangian device such as wave buoys faces inherent limitations for measuring steep waves that are typically found in the surf zone owing to the simultaneous vertical and horizontal motion of the buoy, the waves in the record tend to look more symmetrical around the mean sea level than they actually are such that non-linear effects cannot be properly investigated with a buoy (e.g. Magnusson et al., 1999; Foristall, 2000). Furthermore, remote sensing techniques, which ability to measure nearshore waves have been demonstrated in a myriad of studies, should be mentioned here.

We agree with you on this comment, and the sentence now reads, "As a complement to the fixed sensors and remote sensing methods, buoys have become a good option for obtaining direct measurements of the surface kinematics in various sea states." on Lines 52-53. We believe that the buoy measurements are not an alternative to fixed instruments but are rather a complement to the fixed instruments and remote sensing methods. A short discussion of remote sensing techniques has also been added on Lines 47-49. Further discussion about the limitations of buoys measuring the non-linearity of nearshore waves, including the suggested citations, is included further down on Lines 71-74 and reads as "While buoys have inherent challenges in measuring nearshore waves, including distortion of surface elevation from accelerometer measurements (Magnusson et al., 1999) and inability to resolve second-order non-linearity (Forristall, 2000), they are the only tool that can be used to obtain direct measurements of the kinematics of the surface and observations of the motion of real buoyant objects in the nearshore."

3. l. 44 "Free drifting buoys…" This assertion should be supported with some appropriate references, furthermore the sentence reads oddly (it looks like a word is missing), I suggest you reword it.

We agree that this sentence was a bit strange, but the point has been reiterated differently and is split between Lines 46-47, which reads, " These fixed sensors generally have robust statistics since they measure continuously in the same location." and Lines 54-56 which read as "Free-drifting buoys tend to move through the surf zone very quickly, other studies have

reported buoys reaching approximately 50 cms−1 as a mean drift velocity, which can be further enhanced by breaking waves and bores (Schmidt et al., 2003; Deike et al., 2017)." The point we hope to make here is that the buoys are drifting and therefore do not collect data in one location as the fixed instruments do. Therefore, the data they collect at a specific location is generally shorter than that of a fixed instrument.

4. l. 51 "… however, they are limited to…" As pointed out above, I would say it is actually an inherent limitations of wave buoys.

As described above, this has been commented on on Lines 71-74. However, this specific sentence is still in the Lines 59-61 manuscript since it refers to the *GPSwaves* processing algorithm, which uses horizontal velocities and an assumption of circular wave orbits to estimate the scalar energy density spectrum. The sentence now reads: "GPS-based wave buoys have been effective at measuring deep water ocean waves; however, they are limited to measuring deep water waves due to an implicit assumption of circular wave orbital motion (Thomson et al., 2018)."

5. Fig. 3. The cap on the bottle seems fairly "standard", have you encountered any problems with the seal?

The cap is standard. Generally, the bottles were robust, though we did add an O-ring to the lid to help keep water out. We found that when the bottles were smashed into the sand from the shore break, they could leak a little. While a few of the bottles leaked during the experiment, most of them were fine. At the end of each day, we also used compressed air and a towel to clean out each of the lids and inspect them to ensure they had the best chance of staying dry during consequent deployments. A sentence has been added to the manuscript to address this on Line 112: "The Nalgene water bottle has a standard lid, and an O-ring was added to the mouth of the bottle to reduce water intrusions."

6. l. 99-107 Is there an SD card to keep a record of the data, if not would it be worthy?

There is an SD card that records all of the raw data. The dataset discussed here is all from the raw data downloaded from the SD card. While the buoys have an iridium modem, it was not utilized for this experiment other than some live time tracking to recover the buoys and all of the raw data. A sentence has been added in Lines 118-119 that reads, "The Raspberry Pi Zero also has an SD card where all raw data is stored and downloaded from when the buoys are recovered."

7. l. 123-124 "For nearshore applications…" After reading section 2.3, I see nothing in the processing method to account for non-linear effects. As pointed out above this is quite an important limitation of the concept of employment of these buoys, could you thus please elaborate a bit more?

We agree that this is an important limitation for the original use of the buoys. Due to changing our perspective to focus more on the Level 1 data than the Level 2, we have not continued with any corrections for nonlinearity in the sea surface elevation time series. We have rather noted the limitations of the buoys on Lines 67-69 as the following: "While buoys have inherent challenges in measuring nearshore waves, including distortion of surface elevation from

accelerometer measurements (Magnusson et al., 1999) and inability to resolve second-order non-linearity (Forristall, 2000), they are the only tool that can be used to obtain direct measurements of the kinematics of the surface."

 have there been any losses, if so it is worth mentioning the rate of lost bottles such as the reader can have a proper idea of what involve such deployment?

During this experiment, 2,187 bottles were deployed (buoys were repeatedly deployed), and only one bottle was lost. The buoys deployed also included what we refer to as 'shepherd' buoys which were in the same hull and ballasted the same as the microSWIFTs but had GPS tracker (Garmin Astro) inside them so we could track the movement of the array in real time for recovery. This has been added to the manuscript on Lines 149-152 and reads as, "To track the buoys in live time, we deployed 'shepherd' buoys which had the same hull and ballast as the microSWIFTs but had a live tracking GPS in them to track the current movement of the buoys as they drifted. Over the course of the experiment, 2,187 buoys were deployed, including the shepherd buoys, and only one was lost."

9. l. 153 "Gaps are rare…" Is it also true for GPS data, as it seems to me it is quite common to find some gaps in GPS wave buoys measurements, presumably associated with waves passing over the buoys?

Any gaps in GPS data we expect to be due to temporary submersion or waves overtopping the buoys as well. However, any noticeable gaps in the GPS data called for the buoy to be removed during the quality control phase, so the data left have little to no gaps in the GPS record. Any small gaps in the GPS signals were filled using linear interpolation following the approach of Schmidt et al. 2003. This description has been added to the Lines 185-188 manuscript: " Gaps in GPS measurements occur due to the buoys being overtopped or plunged underwater. These gaps are filled using linear interpolation but are generally minor. Schmidt et al. (2003) used similar GPS-based drifters and found 95% data return rates seaward of the surf zone and 75% data return rates within the surf zone but found linear interpolation was an appropriate method to fill the data gaps, so this methodology is followed."

10. l. 201-202 "we use data when an individual microSWIFT…" Why not considering measurements from buoys inside a circle, centered on AWAC location, with a given radius, according to the bathymetry constraint? Here you might have considered data from buoys at quite different locations along the isobaths.

As part of a compromise with other researchers during DUNEX, we agreed to deploy buoys on the pier's south side primarily; therefore, measurements were limited near the 4.5 meter AWAC. Following another reviewer's suggestion, we have instead computed the significant wave height estimate from the aggregate of all wave realizations outside an approximate surf zone. Changing this analysis led to more measurements across all the missions rather than being constrained to just wave realizations near the 4.5 meter isobath. The analysis results are shown in Figure 9 in the revised manuscript and below. Here, we also include your suggestion as a dark ring around points where the centroid location that the aggregate measurements are within a 300 meter radius. This analysis shows that only 9 of the 63 significant wave height

measurements have an average location within a 300 meter radius of the 4.5 meter AWAC. We appreciate this suggestion and think it is useful, so it is combined with the other reviewer's suggestion as a marker on the comparison scatterplot.

[Figure]

Figure 9. Comparison of the estimated significant wave heights from the microSWIFT arrays, 6-meter AWAC, and 8-meter pressure sensor array (6-meter AWAC and 8-meter array have been corrected for shoaling) to the estimates from the 4.5 m AWAC. While the microSWIFT arrays are not in the same water depth as the 4.5 m AWAC, we see that the microSWIFT arrays characterize the size of the waves with good comparison to the 4.5 m AWAC. The gray bars indicate 95% confidence intervals around each of the significant wave height estimates, computed using a bootstrap method from the distributions of wave heights. The colors of the estimates depict if the microSWIFT array is in the 'shadow' of the pier where we expect a reduction in wave energy.

11. l. 204-207 "The spectra are computed…" I get a bit lost here with the estimate of the number of DOF and the resulting spectral resolution. My understanding is that the 10 min (600 sec) records are divided into three overlapping windows (Nw=3) of 300 sec with a 50% overlap, and then, the average spectrum is band-averaged on five frequency bins (M=5). The number of DOF could be thus roughly estimated as: DOF = 2*Ns*(M+1)/2 = 18, as opposed to 51. Could you detail a bit more the way spectra are computed?

A detailed description of how the degrees of freedom are computed is included in Lines 241-251. The discussion follows: "The microSWIFT spectra are computed using Welch's method, with Hanning windows and 50% overlap between adjacent windows. The energy in each five adjacent frequencies is band-averaged to improve the statistical robustness of each estimate. The equivalent degrees of freedom for each spectrum is computed using the formulation in equation 2 for Hanning windows
from Thomson and Emery (2014).

$$DOF = (8/3)\frac{N}{M}$$    (Equation 2)

Here, N is the number of data points in the time series, and M is the half-width of the window in the time domain. For these spectra, N = 7200, which is 10 minutes (600 seconds) sampling at 12 Hz frequency, and M = 1800, which is the half-width of a single window. After band-averaging the five adjacent estimates, this results in approximately 53 degrees of freedom(rounded to the closest integer). The AWAC measurements consist of a 34-minute record with a sample rate of 2 Hz, and spectra are computed with 13 50%-overlapping windows (512 points per window) leading to approximately 42 degrees, comparable to that of the microSWIFTs (Christou et al., 2011)."

12. l. 210-211 and l.247-248 I am not convinced by the robustness of the spectral analysis. As pointed out in the two comments above I have some doubts on the way spectra are computed. Furthermore, the spectra show discrepancies that might result in quite important differences on bulk parameters, maybe not Hm0, but what about mean periods? These statements should be tone down, I would rather speak of a relatively good qualitative agreement.

The spectral analysis has been toned down to now show an example comparison with the 4.5 meter AWAC since very few measurements had 10 minutes of data along comparable isobaths. Following another reviewer's suggestion, the spectra from each individual microSWIFT are averaged together, and the significant wave height is computed by integrating the averaged spectra and the spectra from the 4.5 meter AWAC. The individual and averaged spectra compare qualitatively well with the 4.5 meter AWAC. The updated figure is shown below. The associated text is now on Lines 256-258: "The qualitative agreement of each microSWIFT spectra and the AWAC suggests that the measurements are useful for further investigating wave properties with the buoys. Future use of these data may investigate the estimation of directional spectra and directional moments, but they are not investigated in this study." Since the paper has been restructured to move away from the focus on Level 2 measurements which, as stated in previous comments, have challenges, we do not compute the mean periods or focus more on bulk wave statistics.

[Figure]

Figure 7. Comparisons of Panel (a) shows the drift tracks of the microSWIFTs from mission 18 plotted over the surveyed bathymetry DEM. Panel (b) shows a subset of the drift tracks where the bathymetry along each track is between -4.3 and -5.3 meters, and each microSWIFT is a different color. Panel (c) shows the spectra computed from a subset of the sea surface elevation time series for each microSWIFT. One error bar is shown for a confidence interval of

the spectra with 53 degrees of freedom. Significant wave heights are computed by numerically integrating the AWAC and averaged microSWIFT spectra.

13. l. 215-218 "Since the microSWIFTs…" In practice how individual waves measured by different buoys are tagged? Could you detail a bit more the "sampling with replacement method"?

In practice, we cannot determine whether two buoys measure the same wave; thus, individual waves are not tagged. Therefore, we define a wave realization as any wave measured by a buoy using a zero-crossing method (data between two adjacent upward zero-crossings in sea surface elevation). Since multiple buoys can measure the same wave as the wave propagates past the buoys, this example wave is included in the aggregate distribution of waves multiple times. We gave an analogy to 'sample with replacement' methods such as monte carlo simulations and bootstrap methods that use a limited dataset and sample with replacement to improve the statistics of the limited dataset. This is what the array of microSWIFTs is doing physically as multiple waves propagate through the array. Further explanation and discussion of this idea is given in Lines 267-273: "Since the microSWIFTs are spatially distributed in the nearshore and sampling simultaneously, some microSWIFTs will measure the same wave as it propagates past multiple buoys. We treat this like a physical 'sampling with replacement' method similar to Monte Carlo or Bootstrap simulation methods known as re-sampling techniques. These re-sampling techniques improve confidence in a statistical estimate from a finite amount of data (Thomson and Emery, 2014). In this case, the finite data is the short period that the buoys sample an area, but multiple datasets from different microSWIFTs, occasionally containing measurements of the same wave, can help improve confidence in the statistics."

14. l. 222-226 and Fig. 8e I am not convinced of the relevance of such an aggregated distribution, as the buoys did not measure the same sea state as they drift; what is the meaning of this significant wave height? The following of the paragraph makes more sense to me. I suggest to remove Fig. 8e and the associated discussion.

The goal of aggregating the wave realization measurements from the buoys is to provide a more robust statistic than just those from a single buoy since they drift quickly through the nearshore. While you are correct, by measuring a larger area of the surf zone and in multiple depths, this significant wave height is not exactly the same as that computed by the AWACs and pressure gauges. We recognize that there is spatial variability in the wave heights, but we maintain that aggregating the wave realizations together should give us a statistical picture of the wave field that should be qualitatively comparable to the fixed sensors. The distribution of wave heights shown in Figure 8e is used just to demonstrate what wave heights were measured during a singular mission and see the contributions from each individual buoy (colors on the histogram correspond to the different colors shown on the drift tracks). The significant wave height that is now shown in Figure 8e is computed only from wave realizations outside of the surf zone. The updated Figure 8 is shown below.

[Figure]

Figure 8. Example of steps in processing each mission. Panel (a) shows the drift tracks of the microSWIFTs from mission 19 plotted over the surveyed bathymetry DEM. Panel (b) shows the same drift tracks as Panel (a) but shows each microSWIFT as a different color. Panel (c) show the time series of computed sea surface elevation, with one-time series being highlighted as an example. Panel (d) is a zoomed-in portion of the overall time series showing the locations of zero crossings and how we define the height of an individual wave in a time series. Panel (e) is the probability density of all wave heights from the entire time series, where the colors show the contribution from each microSWIFT with the corresponding color. The probability density distribution fits a Rayleigh distribution. The vertical line shows the computed significant wave height for this distribution and the 95% confidence interval of the estimate.

15. l.233-247 Did you processed AWAC measurements the same way, i.e. zero-crossing processing method using AST measurements or did you consider the spectral estimates of the significant wave height? For sake of clarity, I suggest you dedicate an appendix to the processing of AWAC measurements

The AWACs are managed by the US Army Corps of Engineers at the Field Research Facility (FRF) where the experiment took place. Since this is the case, we did not do any processing of

the AWAC measurements, rather, we just used the data products that they publish on their data portal (https://frfdataportal.erdc.dren.mil/). This is clarified in the manuscript in Lines 100-102: "The data from these sensors are processed by the FRF staff to produce estimates of the bulk parameters of significant wave height, mean wave period, and mean wave direction for the duration of the field experiment (Figure 2) along with many other wave and current data products." It is also discussed in Lines 244-249 as the following: "The AWAC measurements consist of a 34-minute record with a sample rate of 2 Hz, and spectra are computed with 13 50%-overlapping windows (512 points per window) leading to approximately 42 degrees, comparable to that of the microSWIFTs (Christou et al., 2011). The staff of the Field Research Facility process the AWAC data following the methods described by Earle et al. (1999). These spectral characteristics result in a frequency resolution of 0.016 Hz. Note, however, only the data products processed by the FRF are used in this study, and no processing of the raw AWAC data is done in this study."

Technical Corrections:
l. 73 "our team" and elsewhere, please avoid the "we" and the "us".
This suggestion is appreciated and this choice of language has been reduced as much as possible within the text. However, we believe this is a stylistic choice and an effective way to write in an active voice, which is generally more engaging for the reader. We are also following the style described in Zinsser 2006. The correction has been taken into account, and this type of language has been reduced in the paper.

l. 84 and below the positions should be given with the appropriate distance unit from the origin of the local frame, I assume meters.
Thank you for catching this; it has been corrected in the manuscript.

l. 93 "mean wave period" are you referring to Tm01 or Tm02 or the mean period issued from waveby-wave analysis? This is in line with my comment 15 above.
In this case, the mean wave period is the inverse of the energy-weighted mean frequency or the inverse frequency moment, Tm-10. This is provided as a standard product within the AWAC and pressure sensor data processing on the FRF data portal. Further description of each data product is provided in the metadata contained within the netCDF files for the data. The data from the 8 meter array of pressure sensors, which we present the mean wave period in Figure 2, can be accessed through this link:
https://chlthredds.erdc.dren.mil/thredds/dodsC/frf/oceanography/waves/8m-array/2021/FRF-ocean_waves_8m-array_202110.nc.html. Since we do not use this data for other analysis than context to the microSWIFT dataset, we have chosen to leave the sentence as the mean wave period for simplicity.

References:

1. Thomson, J., Girton, J. B., Jha, R., and Trapani, A.: Measurements of Directional Wave Spectra and Wind Stress from a Wave Glider Autonomous Surface Vehicle, Journal of

Atmospheric and Oceanic Technology, 35, 347–363,
https://doi.org/10.1175/JTECH-D-17-0091.1, 2018.

2. Thomson, R. E. and Emery, W. J.: Data analysis methods in physical oceanography, Newnes, 2014.

3. Zinsser, William. "On writing well: The classic guide to writing nonfiction." New York, NY (2006).

---

## Author Response (AR2)

Response to Reviewers for Minor Revisions on *Measurements of Nearshore Ocean-Surface Kinematics through Coherent Arrays of Free-Drifting Buoys*

Reviewer Comments in Red
Author Responses in Black

Dear Reviewer #2 and Dr. Marc Pezerat,

Thank you for reviewing this manuscript again, and we appreciate your feedback. Each of your comments is addressed below, and we hope that this will further improve the manuscript.

Reviewer #2:

Overview & Big Picture:
I appreciate the authors changes made to the manuscript based on my and others reviews comments. I believe the manuscript is improved by zooming out a bit and not attempting to provide a rigorous comparison between microSwifts and current meter wave statistics. The addition of the thought provoking Section 4 also adds to the manuscript. Although, I believe the manuscript is improved, and should be published after addressing a few concerns (outlined below) one could argue that less rigor results in a weaker manuscript. For instance, the most rigorous investigation into the quality of the microSwift data centers around Fig 9 and lines 270-297. At line 297, the authors conclude, "The agreement in significant wave height and scalar energy density spectra supports that the Level 2 data are useful for investigating wave spectra and statistics."
This is the authors main scientific finding: for some journals, this would not be enough to warrant publication, but I leave it to the editor of this journal to decide whether this is enough for this journal (as I'm not that familiar with this journal).

We agree that this finding is not a major scientific result; rather, we are presenting a dataset to the community. We are following the mission of this journal, which includes the following, "Earth System Science Data (ESSD) is an international, interdisciplinary journal for the publication of articles on original research data (sets), furthering the reuse of high-quality data of benefit to Earth system sciences. The editors encourage submissions on original data or data collections which are of sufficient quality and have the potential to contribute to these aims.

Articles in the data section may pertain to the planning, instrumentation, and execution of experiments or collection of data. Any interpretation of data is outside the scope of regular articles. Articles on methods describe nontrivial statistical and other methods employed (e.g. to filter, normalize, or convert raw data to primary published data) as well as nontrivial instrumentation or operational methods. Any comparison to other methods is beyond the scope of regular articles."

Following this mission statement, we focus on the methods used to develop the instruments (microSWIFTs), the field experiment and data collection, the methods in which these data are processed, and how they are organized for future use. We plan to use these data in further

studies that will specifically focus on scientific results from the dataset and will not include all of the details of the instruments and dataset as a whole. With this in mind, we believe that the findings presented in this study are appropriate for publication in this journal.

"Bigger Points"
This leads me to bigger point 1 that concerns Fig 9. Note that, "agreement in significant wave height" is based on Fig 9, where Hsig_mS vs Hsig_AWAC is scattered. The best fit line of the data is Hsig_mS = 0.61 Hsig_AWAC. This is quite poor agreement in my opinion, especially as the rms error is .37 m, a large fraction of the average Hsig (approximate 1.75)! The smaller Hsig_mS is attributed to shadowing by the pier. What is the slope if the shadowed values are omitted (gray dots in Fig 9)? The assertion that shadowing gives rise to the <1 slope should be tested. Shadowing can also be tested by investigating whether proximity to the pier matters (by coloring the dots in Fig 9 by distance to the pier?). Also, does it matter whether the waves are from the north or the south and the resulting direction of the mS? This should be explored in more detail and differences between Hsig_AWAC and Hsig_mS explored in more detail.
Shadowing only occurs when waves come from an oblique angle, and the microSWIFTs are on the opposing side of the pier from the direction of the waves. Therefore, the metric that we present in Figure 9 includes an investigation of the proximity to the pier (within the shadow must be within 200 meters of the pier) as described in lines 300-302, "Being in the pier 'shadow' is defined here as missions when the average location of the microSWIFTs during a mission is within 200 meters of the pier, and waves are coming from the other side of the pier based on the mean wave direction from the 8-meter array (furthest offshore sensor)."

If the shadowed values are omitted, the slope of the linear regression is 0.53, which suggests worse agreement; however, omitting all of the shadowed points reduces our sample size to 71% of the available data, which could lead to weaker statistics. We also do not expect perfect agreement between the microSWIFT arrays and the AWAC measurements since they measure at different locations, and the microSWIFT arrays measure waves in many different depths since we use data only outside of the approximated surf zone. The comparison should rather capture the general trends, microSWIFTs measure large waves in large wave conditions and small waves in small wave conditions, primarily what we see in this comparison. Further offshore, the microSWIFTs have shown much stronger agreement with nearby instruments for the significant wave height, as shown in Figure 5 of Thomson et al. 2023 (In Review, [https://github.com/SASlabgroup/SWIFT-codes/blob/master/Documents/microSWIFTs_CEJrevision_6Sep2023.pdf](https://github.com/SASlabgroup/SWIFT-codes/blob/master/Documents/microSWIFTs_CEJrevision_6Sep2023.pdf)).

The authors also state that the difference is due to a short time series. I suspect not, as addressed later in this review. Also, the AWAC is in about 4.5 m of water, which according to gamma=0.35 in the manuscripts means that waves bigger than 1.5 m are breaking at this AWAC. I think only Hsig when BOTH instruments are outside the surfzone should be compared. Hsig_mS in a region of breaking waves will not be reliable (as mentioned by the authors) as mS surf broken waves. Thus, the "good" part of the plot, Hsig_AWAC < 2 m, the relationship between mS and 4.5 m AWAC isn't very good. It doesn't make sense to include data when the mS might be surfing. If the mS might be surfing, that data should not be included in this

comparison. Unfortunately, the more I stare at Fig 9, the more I'm not so sure that microSwifts can tell me anything accurate about wave heights. If they can't get this statistic really well, what does that mean for other higher order statistics?

The choice of gamma=0.35 is a conservative estimate for gamma, suggesting that the surf zone may be narrower than our surf one edge estimate, and the location of the 4.5 meter AWAC should still be in a zone of intermittent breaking even under larger wave conditions.  This was confirmed by visual observations during data collection.   The caption associated with Figure 9 is now revised as , "Comparison of the estimated significant wave heights from the microSWIFT arrays, 6-meter AWAC, and 8-meter pressure sensor array (6-meter AWAC and 8-meter array have been corrected for shoaling) to the estimates from the 4.5 m AWAC. While the microSWIFT arrays are not in the same water depth as the 4.5 m AWAC, we see that the microSWIFT values are similar to the 4.5 m AWAC values. The gray bars indicate 95% confidence intervals around each of the significant wave height estimates, computed using a bootstrap method from the distributions of wave heights. The colors of the estimates depict if the microSWIFT array is in the 'shadow' of the pier, where we expect a reduction in wave energy. For significant wave heights greater than 2 meters, intermittent breaking may be occurring at the 4.5 meter isobath, leading to worse agreement between the AWAC and microSWIFT measurements."

We agree that the data from the microSWIFTs may not be well-suited to investigating higher-order statistics. This is why the manuscript has been restructured to focus on the kinematics of the surface and what we can learn from the Level 1 data with examples given in section 4 of the manuscript, including investigating 'surfing' transport of buoyant objects, surface kinematics, and spatial variability of breaking waves. We do not claim that the microSWIFT arrays are a precise tool for measuring wave statistics. Instead, we suggest that microSWIFT wave heights are useful contextual data when investigating breaking/surfing kinematics using the raw motion data.

Although this doesn't affect the quality of the paper, because the spectra are only used qualitatively, I still contend that the EFD (effective degrees of freedom) isn't correct for the spectra calculated in this MS. And it should be done correctly. First, notice that in Fig 7 c, the variability of each spectral estimate is bigger than the 95% bars. This means that each peak in the spectra is real, which I doubt. I believe that the 95% bar should be longer more consistent with the variability within the spectra. I.e. the EDF used is too large. The authors state that for the microSwift the EDF is given by equation (2) in the MS,

EDF = (8/3) N/M

(from Thomson and Emery, 2014 table 5.5, but this is actually from Priestley 1981) where N is the number of points in the time series, and M is the 1/2 width. They use N=7200 (600 s x 12 Hz) and M=1800. Then 5 frequencies are averaged. So

EDF = (8/3) * (7200/1800) * 5 = 53. I don't think (8/3) N/M is being correctly used. Either that, or the formula itself is incorrect. I'm not sure, because these formula in T&E are not derived so it isn't clear where N and M come from. Regardless, for a spectra with 3 non overlapping blocks of data, the dof=6 and overlapping the blocks of data reduces the number of degrees of freedom. Thus for 3 blocks of data, the MAXIMUM EDF = 6 and then averaging 5 frequencies would yield

30 dof. The authors can not have more than 30 dof. T&E hint at this on page 476... " Spectra are then computed for each of the K segments and the spectral values for each frequency band then block averaged to form the final spectral estimates for each frequency band. If there is no overlap between segments, the resulting DoF for the composite spectrum will be 2K. This assumes that the individual sample spectra have not been windowed and that each spectral estimate is a chi-squared variable with two DoF."

I believe T&E can be confusing, especially table 5.5. EDF are considered in a variety of places. EDFs are derived in
http://pordlabs.ucsd.edu/sgille/sioc221a/lecture11_notes.pdf
and I highly suggest looking at this doc. Here, it is clearly shown that the EDF only depends on the number of blocks (aka segments or chunks ==Nb, "K" in T&E) averaged to make the spectra (which does not depend on N the number of samples within the block). With no overlap, EDF = 2*Nb (as outlined above), but since there is overlap, and a Hanning window is used, 2 becomes 1.9 and
EDF = 1.9 * Nb = 1.9 * 3 = 5.7
but 5 frequencies are averaged so EDF = 5.7 * 5 ~= 29.
The 53 stated in the MS is not the number of degrees of freedom for this spectra. Note, if N/M = 2, i.e. the window is 1/2 the length of the entire time series, which it is, then 8/3 *2 = 5.33 which is similar to the 5.7 above. Also note that in the above linked pdf it is stated that, "So what of the other texts? The 2014 edition of Thomson and Emery is as misleading as the earlier editions."
I believe this is in reference specifically to Table 5.5 of T&E, so according to Gille, who I trust, maybe table 5.5, where the 8/3 N/M comes from, isn't the best reference regarding spectra dof? However, T&E state, "Nuttall and Carter (1980) report that 92% of the maximum number of equivalent degrees of freedom (EDoF) can be achieved for a Hanning window, which uses 50% overlap." I.e. 6 becomes 6*.92 = 5.52, and 5.52*5 = 28 dof. One less than the Gille formula.

For the AWAC, assuming a Hanning window,
EDF = 1.9 * 13 ~= 25 dof
not 42. Again, the larges EDF for the AWAC is 2*13 = 26. but the overlapping blocks result in slightly less.

This has been corrected. The following shows the adjusted Figure 7 with the equivalent degrees of freedom for the microSWIFTs equal to 28 and the equivalent degrees of freedom for the AWAC equal to 25. Lines 242-252 have been adjusted accordingly and are now the following. "The microSWIFT spectra are computed using Welch's method, with five-minute (3600 sample) Hanning windows and 50% overlap between adjacent windows. The energy in each of the five adjacent frequencies is band-averaged to improve the statistical robustness of each estimate. The equivalent degrees of freedom for each microSWIFT spectrum is 28. This is based on the ten-minute time series (7200 samples at a 12 Hz sampling rate) used for each spectral estimate with three five-minute windows (50% overlap). Each window contributes 2 degrees-of-freedom and band-averaging the five adjacent frequencies increases the effective degrees of freedom by a factor of five. Due to the 50% overlap of the Hanning windows, the equivalent degrees of freedom are reduced to 92% of the maximum degrees of freedom (Nuttall and Carter, 1980). Therefore, the equivalent degrees-of-freedom for the microSWIFT spectra is 28 (3 windows * 2

degrees-of-freedom * 5 frequency bands * 0.92 = 28). The AWAC measurements consist of a 34-minute record with a sample rate of 2 Hz, and spectra are computed with 13 50%-overlapping windows (512 points per window) and no band-averaging, leading to approximately 25 degrees-of-freedom, comparable to that of the microSWIFTs (Christou et al., 2011)."

[Figure]

"Smaller Points"
Line 290: "We also expect that the microSWIFT arrays may under-predict some significant wave heights as the sampling windows are shorter than the AWAC, potentially not measuring the largest and least likely waves in the distribution and times that the microSWIFTs are within the 'shadow' of the pier."
This seems a bit misleading. The shortness of the time series doesn't bias the difference between mS and AWAC Hsig, it just creates more variability. The authors could have also said,

"We also expect that the microSWIFT arrays may over-predict some significant wave heights as the sampling windows are shorter than the AWAC, potentially over representing the largest and least likely waves in the distribution and not measuring enough of the smaller waves."
This has been corrected. Lines 297-300 are now the following. "We also expect that the microSWIFT arrays have more variability in their significant wave height estimates since the sampling windows are shorter than the AWAC, potentially over-representing or under-representing the largest and least likely waves in the distribution. Further underestimation could be due to the microSWIFTs being within the 'shadow' of the pier."

Line 323: Fig 11a. Is this GPS u? Or the Kalman filtered u? Hopefully the Kalman filtered u. If GPS velocities, are they de-spiked? Fig 11b. Is the acceleration in the vertical reference frame? If not it should be as the "body frame of reference" is not as obviously useful. The Kalman filtered velocities and accelerations should be used in this figure.
The velocities shown in Figure 11a are GPS velocities that have been de-spiked. The GPS velocities are already measured in the Earth reference frame, so we chose to use these velocities with less associated processing. Additionally, Figure 11b shows the vertical accelerations in the "body frame of reference" and have also been de-spiked. Again, following the same argument as for the GPS velocities, this is a less processed measurement (i.e., Level 1 data). These choices align with the reframing of the paper as a whole to focus more on the lightly processed measurements and what can be learned from them.

Line 273. 1.416 should be 1.414 as it is 2^{1/2}.
This has been corrected. Lines 279-280 are now the following. "The significant wave height is computed by first computing the root-mean-square of the wave heights and then multiplying by a factor of 1.414 to convert to significant wave height for a Rayleigh distribution (Dean and Dalrymple, 1991)."

Also, does your boot strapping method yield
Hs [1 - 0.41 / N^(1/2) ] < Hs < Hs [1 + 0.41/N^(1/2)]
for the 95% confidence limits? where N is the number of waves in the estimate. I believe these are the 95% confidence limits of Hs based on a Rayleigh distribution and should be confirmed by bootstapping. In my opinion, it is better to use a derivable formula than just say, "we got this number by bootstrapping" as there is no way for a reader to determine if that number is correct.
The boot-strapping method does not exactly yield these confidence intervals, though we agree that they should converge as the square root of the number of realizations.   We show the alternative version of Figure 9 below, using the equation from the reviewer.   These intervals are smaller than the boot-strap method, presumably because the empirical boot-strapping includes other sources of uncertainty inherent to the data.   We prefer to use these wider intervals, rather than rely purely upon the theoretical Rayleigh distribution (though the shape does seem to match our aggregate data, as shown in Figure 8e).   In either case, the statistical uncertainty in each Hs estimate from the microSWIFT buoy arrays is insufficient to fully account for the mis-match with the AWAC observations.  Again, we attribute these differences to the lack of colocation and alongshore variability.

[Figure]

Dr. Marc Pezerat Comments:

l.5 "the buoy's global position" -> why global ?
Here, we wanted to be precise that the positions are from the GPS receiver and are, therefore, global positions. However, this may be unnecessary to specify, so it has been removed from the manuscript, and the line is now the following. "The microSWIFT is a small buoy equipped with a GPS module to measure the buoy's position and horizontal velocities and Inertial Measurement Unit (IMU) to directly measure the buoy's rotation rates, accelerations, and heading."

l.5 and 6 "horizontal velocities and accelerations" -> You should precise that you consider horizontal velocities obtained from GPS measurements not integrated from accelerations (according to l. 319).
This has been corrected in conjunction with the previous correction. See the comment above.

l.32 to 40 Why referring to modelling, it is a bit misleading, as beyond the scope of this study.
This section is included as a transition between the background and how the dataset can be used in the future. We believe there are still many questions regarding how phase-resolved wave processes are included in phase-averaged models and that the data presented in this paper may be useful in answering those questions. While this is not a modeling study, this section is a useful primer for where this data can be used in the future.

Somewhere around l. 50 It would be nice to distinguish free-drifting and moored buoy's measurements.
A further distinction between free-drifting and moored buoy's measurements was added between lines 49-55. The adjusted lines are the following. "As a complement to the fixed sensors and remote sensing methods, wave buoys are another option for obtaining direct measurements of the surface kinematics in various sea states. Wave buoys can be either free-drifting or moored. Moored buoys are effectively Eulerian wave measurements, with some movement due to the scope of the mooring, while free-drifting wave buoys are closer to Lagrangian measurements but move as a result of the wind, currents, wave-induced drift (Stokes drift), and surfing on broken waves. Free drifting buoys are essential for understanding how buoyant objects move in the nearshore (Spydell et al., 2007; Schmidt et al., 2003). Free-drifting buoys tend to move through the surf zone very quickly; prior studies have reported buoys reaching approximately 50 cm s−1 as a mean drift velocity."

l. 68-69 "they are the only tool…" -> not the only one.
This has been corrected, and the sentence is now the following. "While buoys have inherent challenges in measuring nearshore waves, including distortion of surface elevation from accelerometer measurements (Magnusson et al., 1999) and inability to resolve second-order non-linearity (Forristall, 2000), they are one of the few tools that can be used to obtain direct measurements of the kinematics of the surface"

l.158-159 "Using this definition of $\gamma_s$ , the variable $H_s$ represents the offshore significant wave height (will use measurements from the 8-meter pressure gauge array, location in Figure 1,

panel (c))" -> Perhaps "… the variable Hs represents the offshore significant wave height, here measured from the 8 m-pressure gauge array…"

This sentence has been adjusted in the manuscript. The line is now the following. "Using this definition of γs, the variable Hs represents the offshore significant wave height, here measured from the 8 m-pressure gauge array (location in Figure 1, panel (c)), and the variable d represents the water depth during the mission."

Section 3. Could you introduce subsections to ease the reading?

Four subsections have been added to section 3. The added subsections are titled Data Cleaning and Level 1 Data, Level 2 Data, Spectral Exploration of microSWIFT Data, and Zero-Crossing Exploration of microSWIFT Data.

l. 271 "The significant wave height is computed from aggregated wave height measurements outside the approximate surf zone" -> So, If I understand correctly Fig. 8e is built from elevation timeseries when buoys are located seaward of the surf zone edge?

This is correct. Figure 8e is built from wave realizations when buoys are located seaward of the surf zone edge estimate. To clarify this in the manuscript, the caption is now the following. "Example of steps in processing each mission. Panel (a) shows the drift tracks of the microSWIFTs from mission 19 plotted over the surveyed bathymetry DEM. Panel (b) shows the same drift tracks as Panel (a) but shows each microSWIFT as a different color. Panel (c) shows the time series of computed sea surface elevation, with one time series being highlighted as an example. Panel (d) is a zoomed-in portion of the overall time series showing the locations of zero crossings and how we define the height of an individual wave in a time series. Panel (e) is the probability density of all wave heights, seaward of the approximate surf zone edge, where the colors show the contribution from each microSWIFT with the corresponding color. The probability density distribution fits a Rayleigh distribution. The vertical line shows the computed significant wave height for this distribution and the 95% confidence interval of the estimate."

l. 306 "These measurements can [help] investigate…"

This has been corrected, and line 306 is now the following: "These measurements can help investigate buoyant particles' cross and along-shore transport under various forcing conditions."